# Single-cell and chromatin accessibility profiling reveals regulatory programs of pathogenic Th2 cells in allergic asthma

Matarr Khan[1], Marlis Alteneder[1], Wolfgang Reiter [2,3], Thomas Krausgruber [4,5], Lina Dobnikar[4], Moritz Madern [1], Monika Waldherr [1,6], Christoph Bock [4,5], Markus Hartl [2,3], Wilfried Ellmeier [1], Johan Henriksson [7] & Nicole Boucheron [1] ✉

Lung pathogenic T helper type 2 (pTh2) cells are important in mediating allergic asthma, but fundamental questions remain regarding their heterogeneity and epigenetic regulation. Here we investigate immune regulation in allergic asthma by single-cell RNA sequencing in mice challenged with house dust mite, in the presence and absence of histone deacetylase 1 (HDAC1) function. Our analyses indicate two distinct highly proinflammatory subsets of lung pTh2 cells and pinpoint thymic stromal lymphopoietin (TSLP) and Tumour Necrosis Factor Receptor Superfamily (TNFRSF) members as important drivers to generate pTh2 cells in vitro. Using our in vitro model, we uncover how signalling via TSLP and a TNFRSF member shapes chromatin accessibility at the type 2 cytokine gene loci by modulating HDAC1 repressive function. In summary, we have generated insights into pTh2 cell biology and establish an in vitro model for investigating pTh2 cells that proves useful for discovering molecular mechanisms involved in pTh2-mediated allergic asthma.

Asthma is a chronic lung disease affecting at least 300 million people worldwide and is among the most common non-communicable diseases globally[1–3]. It is characterised by increased airway inflammation, bronchial hyperresponsiveness and airway remodelling; leading to shortness of breath, chest tightness and wheezing[1]. Allergic asthma, the most frequent form of asthma in children and adults[1,2,4], is orchestrated by type 2 cytokine-producing cells such as type 2 innate lymphoid cells and T helper type 2 (Th2) cells along with epithelial-derived cytokines[1]. Upon allergen exposure, airway epithelial cells release a plethora of molecules including proinflammatory cytokines and alarmins[5,6]. The latter include Interleukin-25 (IL-25), IL-33 and thymic stromal lymphopoietin (TSLP)[7], which in concert with antigen-presenting cells such as dendritic cells, promote the differentiation of Th2 cells from naïve CD4+ T cells resulting in the secretion of type 2 cytokines, including IL-4, IL-5, IL-9 and IL-13, that drive disease progression[8].

Studies in humans and mice have identified a Th2 cell subset termed pathogenic Th2 (pTh2) cells as the major inducers of allergic inflammation due to their propensity to secrete high levels of the type 2 cytokines IL-13 and particularly IL-5 (refs. 9–18). Due to their proallergic phenotype, pTh2 cells have become an important drug target not only in allergic asthma but also in other allergic diseases including

[1]Medical University of Vienna, Center of Pathophysiology, Infectiology and Immunology, Institute of Immunology, Division of Immunobiology, Vienna, Austria. [2]Max Perutz Labs, Mass Spectrometry Facility, Vienna Biocenter Campus (VBC), Vienna, Austria. [3]University of Vienna, Center for Molecular Biology, Department of Biochemistry and Cell Biology, Vienna, Austria. [4]CeMM Research Center for Molecular Medicine of the Austrian Academy of Sciences, Vienna, Austria. [5]Medical University of Vienna, Center for Medical Data Science, Institute of Artificial Intelligence, Vienna, Austria. [6]FH Campus Wien, Department of Applied Life Sciences/Bioengineering/Bioinformatics, Vienna, Austria. [7]Umeå University, Umeå Centre for Microbial Research (UCMR), Integrated Science Lab (Icelab), Department of Molecular Biology, Umeå, Sweden. ✉e-mail: nicole.boucheron@meduniwien.ac.at

allergic rhinitis, atopic dermatitis, eosinophilic esophagitis and food allergy[12,19–22]. Despite the central role of pTh2 cells in allergic asthma, the cellular and molecular processes important for their differentiation and effector function have not been fully dissected[23–25]. A major obstacle to our understanding of pTh2 cell regulation is the lack of an in vitro model allowing high-throughput molecular and functional characterisation, which could potentially pave the way for developing novel therapies targeting them in allergic diseases.

It is well established that the differentiation of naïve CD4[+] T cells into distinct effector T helper (Th) subsets is controlled by multiple factors such as the cytokine milieu and chromatin state of the cells[26]. The latter has a profound effect on the acquisition, stability and function of Th subsets[27]. The chromatin state of a cell is regulated by various epigenetic mechanisms such as reversible lysine acetylation mediated by histone acetyltransferases and histone deacetylases (HDACs)[28,29]. We and others have demonstrated critical roles for HDACs in T cell biology[30–32].

HDACs have garnered interest as potential therapeutic targets in allergic asthma[33–35]. More recently, natural HDAC inhibitors such as short-chain fatty acids (SCFAs) have been considered for potential therapeutics for allergic asthma due to their immunomodulatory properties, mainly by suppressing the enzymatic activity of HDACs[36,37]. Although promising, dissecting the role of HDACs in allergic asthma and particularly on pTh2 cell differentiation is needed to understand whether suppressing the activity of HDACs either pharmacologically or with natural inhibitors might represent a therapeutic approach to limiting allergic asthma. We previously reported that the deletion of HDAC1, a member of the class I histone deacetylases[28,29] in T cells potentiated ovalbumin-induced allergic asthma[38]. However, the role of HDAC1 in pTh2 cell differentiation and effector function in response to the most important indoor aeroallergen for humans, house dust mite (HDM), has not been defined.

Here, using IL-13 tdTomato reporter mice[39] along with a HDM airway inflammation model and single-cell RNA sequencing (scRNA-seq) of CD4[+] T cells from lungs of healthy and diseased wild type (WT) and HDAC1 conditional knockout (HDAC1-cKO) mice, we investigate the regulation of mouse lung pTh2 cells, including their heterogeneity and subset-specific gene signatures. We identify and validate the presence of two distinct subsets of lung pTh2 cells, which express a plethora of proinflammatory molecules including *Il4*, *Il5*, *Il13*, *Tnfsf11*, *Areg*, *Tgfb1*, *Calca* and *Furin*, and exhibit shared and distinct transcriptional signatures. We observe some overlap between the pTh2 cell subsets and other lung ST2[+] Th cells and based on our data, establish a flow cytometry strategy to distinguish them. From our scRNA-seq analysis, we identify conditions that promote the generation of pTh2 in vitro thus resolving a major obstacle in allergy research. Our data establish an important function for HDAC1 in regulating the differentiation and pathogenicity of lung and in vitro generated pTh2 cells. Overall, our findings delineate the heterogeneity and signatures of pTh2 cells and generate insights into their molecular regulation.

## Results

### HDAC1 is essential to restrict HDM-induced airway inflammation

We have previously reported that the ablation of HDAC1 in T cells enhanced ovalbumin-induced allergic asthma[38]. But the role of HDAC1 in response to HDM, an environmental allergen, is unknown. To determine the impact of HDAC1 deletion in HDM-induced allergic asthma, we sensitised and challenged WT and HDAC1-cKO mice with HDM (Supplementary Fig. 1a). Our analyses revealed an increased influx of eosinophils (SiglecF[+]CD11c[-] cells) (Supplementary Fig. 1b–d) and total cellular infiltration (Supplementary Fig. 1e) in the bronchoalveolar lavage (BAL) of HDAC1-cKO mice, indicative of enhanced airway inflammation. To assess the role of HDAC1 in pathogenic type 2 cytokine expression by lung Th cells, we isolated lung cells from phosphate buffer saline (PBS) and HDM-exposed mice and

restimulated them with PMA and ionomycin. Loss of HDAC1 augmented IL-5 and IL-13 expression in the ex vivo restimulated Th cells (Supplementary Fig. 1f, g). These results highlight the importance of HDAC1 in restricting HDM-induced airway inflammation and as a potential regulator of pTh2 cell differentiation.

### scRNA-seq analysis of lung CD4[+] T cells uncovers the heterogeneity of pTh2 cells in response to HDM

Studies have identified pTh2 cells as the main drivers of allergic inflammation[9–18], however, our understanding of the heterogeneity, signatures, and regulation of pTh2 cells is incomplete. To profile lung pTh2 cells and to dissect the role of HDAC1 in these cells, we performed scRNA-seq of lung CD4[+] T cells from healthy and diseased mice and used IL-13 tdTomato-reporter mice to enable us to track all IL-13-producing (IL-13[+]) Th cells in response to HDM, including pTh2 cells[18]. We isolated lung cells and sorted IL-13[+] Th cells, non-IL-13 producing (IL-13[-]) Th cells, and naïve CD4[+] T cells (naïve) from the lungs of WT and HDAC1-cKO mice exposed to PBS or HDM (as in Supplementary Fig. 1a). We obtained a total of ten samples based on the two genotypes (WT and HDAC1-cKO) and experimental conditions (PBS and HDM) (Fig. 1a).

After quality control, integration and unsupervised clustering of our data, we identified fourteen distinct clusters (Supplementary Fig. 2a). We used a combination of known receptors, transcription factors (TFs) and cytokines to annotate the clusters (Supplementary Fig. 2b–d). Clusters 9, 10, 11, 12, and 13 were excluded from subsequent analyses due to cells of non-Th cell origin (clusters 11 and 13) or a small number of cells (Supplementary Fig. 2b and Supplementary Data 1). Our single-cell analysis revealed the presence of different CD4[+] T cell populations in the lung (Fig. 1b). Cells in clusters 0, 1 and 7 originated from naïve cells and expressed *Klf2*, *Ccr7* and *Sell* (CD62L). Clusters 0 and 1 differed mostly in the expression of some ribosomal genes, while cells in cluster 7 showed upregulation of an interferon signature. Cells in cluster 3 represent activated cells with enhanced expression of *Cxcr3*, *Trat1 and S1pr1*. Cells in clusters 4 and 8 expressed signatures of Th17 (*Il17a*, *Ccr6* and *Rorc*) and Th1 (*Ccl5*, *Cxcr3* and *Tbx21*) cells, respectively (Fig. 1c, Supplementary Fig. 3 and Supplementary Data 2).

Interestingly, both clusters 2 and 5 contained cells expressing the pTh2 genes *Il4*, *Il5* and *Il13*, and additionally expressed *Il10*, *Areg*, *Tnfsf11*, *Calca* and *Furin*. Distinguishing clusters 2 and 5 are the expression levels of pathogenic type 2 molecules and the tissue retention marker, *Cd69*. Cells in cluster 2, which we termed pathogenic effector Th2 (peTh2) cells, expressed higher levels of *Il4*, *Il5* and *Il13*, while cells in cluster 5 showed enhanced expression of *Cd69* (Fig. 1c, Supplementary Fig. 3 and Supplementary Data 2). We named the cells in cluster 5 as Th2 tissue-resident memory (Th2 Trm) cells owing to their high mRNA expression of *Cd69*, a marker gene for Trm cells[40]. Moreover, previous studies have demonstrated that Th2 Trm cells emerge in the lungs of mice within a week following allergen exposure, and persist for months[41,42], thus supporting our classification of cells in cluster 5 as Th2 Trm cells.

Next, we performed gene set enrichment analysis (GSEA) to determine the similarity of the pTh2 cell subsets we have identified with recently reported airway pTh2 cells[18]. We used the peTh2 and Th2 Trm gene sets (Supplementary Data 2) and compared them with the recently published airway pTh2 gene set[18]. We found that both the peTh2 (Fig. 1d) and Th2 Trm (Fig. 1e) cell subsets were highly enriched for genes from the published gene set. The previous study did not further investigate whether the airway pTh2 cells contained distinct pTh2 cell subsets, however, our analysis indicates a strong overlap between these cells and the peTh2 and Th2 Trm cell subsets from this study (Fig. 1d–f and Supplementary Data 3), indicating that the published gene set might contain gene signatures belonging to both peTh2 and Th2 Trm cells. These observations highlight the strength of our approach of using IL-13 reporter mice, as it enables us to identify

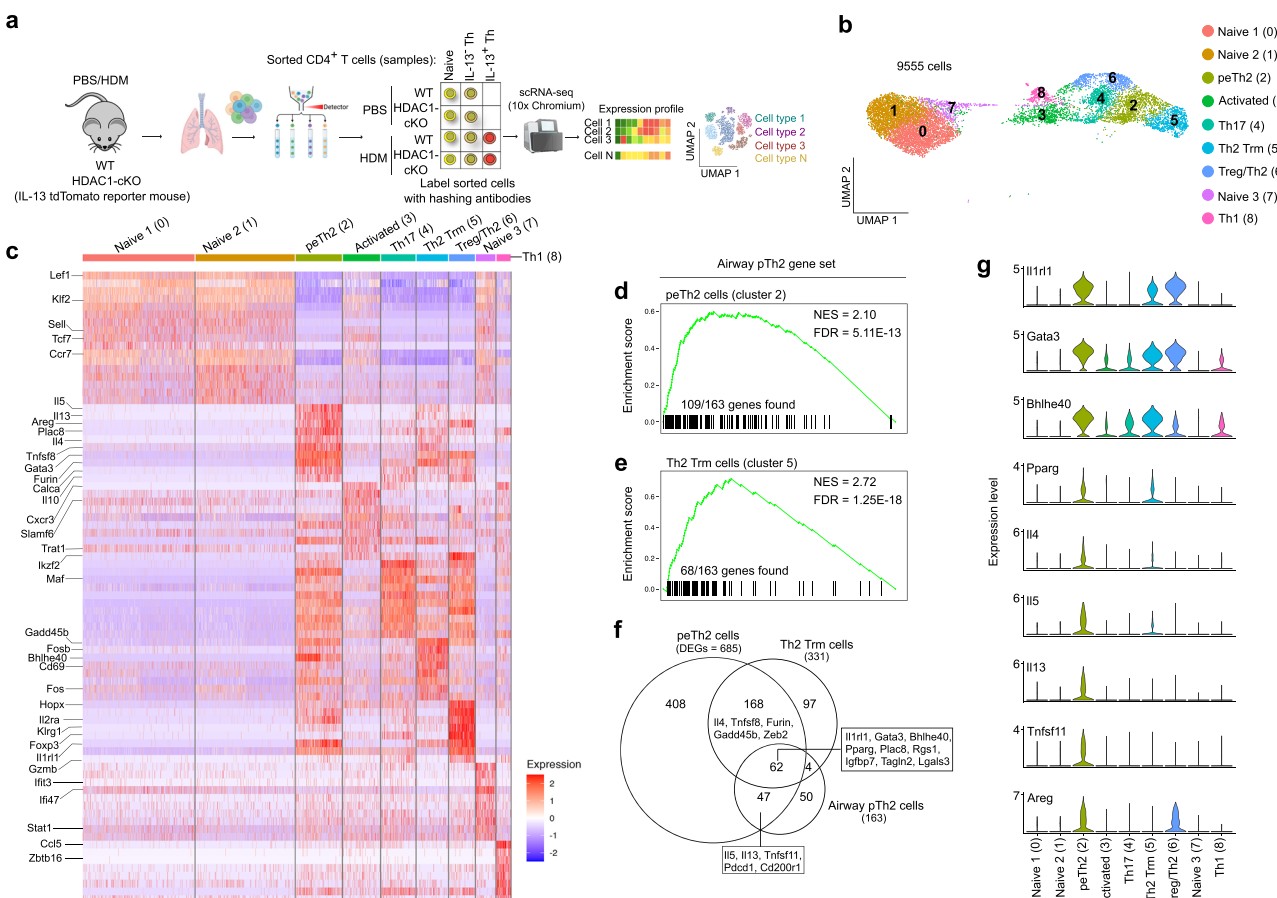

**Fig. 1 | scRNA-seq analysis of lung CD4⁺ T cells uncovers the heterogeneity of pTh2 cells in response to HDM. a** Experimental design for scRNA-seq analysis of lung CD4⁺ T cells. Partly created in BioRender. Boucheron, N. (2025) https://BioRender.com/f94i654. After obtaining single-cell suspensions from the lungs of WT (HDAC1^f/f x CD4-Cre^-/-) and HDAC1-cKO (HDAC1^f/f x CD4-Cre^+/+) IL-13 tdTomato-reporter mice that were sensitised and challenged with PBS or HDM (as in Supplementary Fig. 1a), we sorted the following lung CD4⁺ T cells: naïve (TCRβ⁺CD4⁺CD62L⁺CD44⁺IL-13⁻), IL-13⁻ Th (TCRβ⁺CD4⁺CD62L⁻CD44⁺IL-13⁻), and IL-13⁺ Th (TCRβ⁺CD4⁺CD62L⁻CD44⁺IL-13⁺) cells. A total of ten samples based on the two genotypes (WT or HDAC1-cKO) and experimental conditions (HDM or PBS) were obtained. Each sample was labelled with a unique hashtag oligonucleotide (HTO). All ten samples were pooled for single-cell RNA-sequencing (scRNA-seq) analysis, and each sample is a pool of cells from three different experimental animals per group. The two independent scRNA-seq experiments were integrated for the analyses. **b** Uniform manifold approximation and projection (UMAP) of the different lung CD4⁺ T cell clusters identified. **c** Heatmap shows the top 10 differentially expressed genes (DEGs) per cluster. **d–f** Comparing the transcriptional

signatures of lung peTh2 and Th2 Trm cell subsets identified in the present study to that of published airway pTh2 cells[18]. **d, e** Enrichment plots depicting the association of lung peTh2 cells (**d**) and Th2 Trm cell (**e**) with airway pTh2 cells. The list of all genes in peTh2 cells (cluster 2) and Th2 Trm cells (cluster 5) were compared with the list of DEGs (adjusted *P*-value < 0.05) in the airway pTh2 gene set. Adjusted *P*-values were calculated using Seurat's default two-tailed Wilcoxon rank sum test with Bonferroni correction. For the enrichment plots in **d, e** *P*-values were calculated using fgsea's adaptive multilevel splitting Monte Carlo scheme with Benjamini-Hochberg correction. **f** Venn diagram depicting the overlap between the DEGs (adjusted *P* value < 0.05; calculated using Seurat's two-tailed Wilcoxon rank sum test with Bonferroni correction) in peTh2 and Th2 Trm cells from this study and the DEGs in the published airway pTh2 cells. **g** Violin plots of selected marker genes associated with lung pTh2 cells. WT wild type, HDAC1-cKO HDAC1-conditional knockout, PBS phosphate-buffered saline, HDM house dust mite, Th T helper, pTh2 pathogenic Th2, peTh2 pathogenic effector Th2, Th2 Trm, pathogenic Th2 Tissue resident memory, FDR false discovery rate, NES normalised enrichment score.

distinct pTh2 cell subsets (that is, peTh2 and Th2 Trm cells) and define their signatures.

Our analysis further uncovered a unique Th subset consisting of both Treg and Th2 signatures (cluster 6), thus we named cells in this cluster as Treg/Th2 cells (Fig. 1b, c and Supplementary Fig. 3). The signature of these Treg/Th2 cells included enhanced expression of *Il1rl1*, *Gata3*, *Bhlhe40*, *Areg*, *Gzmb* and *FoxP3*, many of which are also enriched in peTh2 and Th2 Trm cells (Fig. 1c, g). Furthermore, in contrast to a previous report[43], we did not identify a distinct *Areg*-expressing pTh2 cell subset lacking *Il5*. Instead, we found co-expression of *Areg* and *Il5* by peTh2 cells (Fig. 1g).

Similar to pTh2 cells, Treg/Th2 cells also expressed amphiregulin (*Areg*) and *Il10*, encoding Areg and IL-10, respectively (Fig. 1c). Both Areg and IL-10 are regarded as anti-inflammatory, tissue-protective and functional "Treg"-associated mediators[44,45]. However, recent

reports indicated context-dependent roles for these mediators. For example, Areg was shown to play a protective role during viral infection by inducing tissue repair[44]. However, Areg secretion by pTh2 cells in response to HDM potentiates lung fibrosis[43]. Also, IL-10 production by HDM-specific T cells augments Th2 cell differentiation and allergic asthma[46]. Therefore, the protective effect of Areg and IL-10 might be dependent on cell type, tissue localisation and cellular interactions.

During allergic responses in the lung, Tregs might get subverted and lose their regulatory functions, which licences them to acquire a Th2 signature and promote allergic airway inflammation[47]. A hallmark of Treg subversion is the expression of the Notch4 receptor[47]. The Treg/Th2 cells we identified do not express Notch4, are Foxp3 positive (Supplementary Data 2) and show high expression of *Rora*, *Ikzf2* and *Il2ra* (Fig. 1c and Supplementary Data 2) suggesting that these cells are stable, tissue adapted Tregs with repressive ability as previously

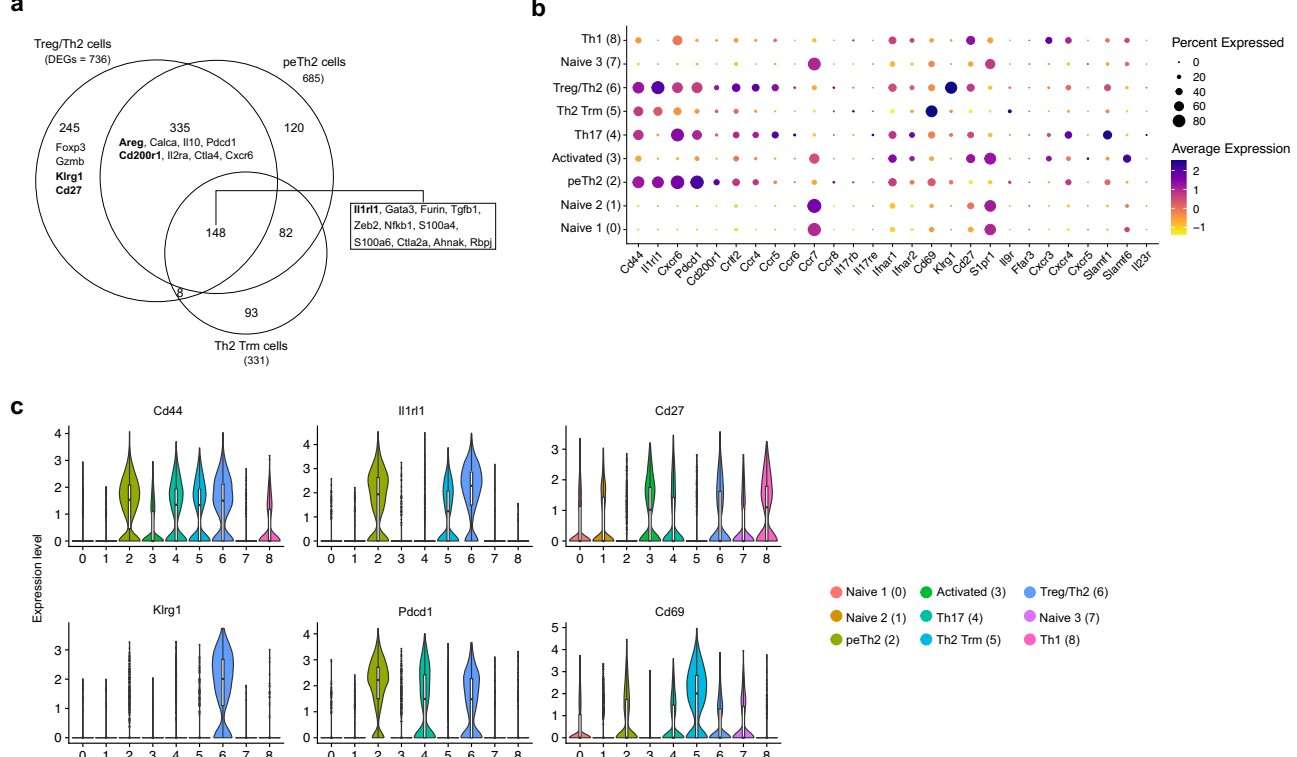

**Fig. 2 | Comparison of lung ST2⁺ Th subsets. a** Venn diagram depicting the overlapping DEGs (adjusted *P*-value < 0.05) for the different lung ST2⁺ Th subsets (peTh2, Th2 Trm, Treg/Th2 and Treg) we have identified. Adjusted *P*-values were calculated using Seurat's default two-tailed Wilcoxon rank sum test with Bonferroni correction. **b** Dot plot showing the expression of selected genes. The size of the dot represents the percentage of cells expressing the indicated gene per cluster and the colour intensity indicates the scaled average expression level of the gene. **c** Violin plots of selected surface markers to distinguish lung ST2⁺ Th subsets. For **c** two

independent experiments with 10 samples each were performed, and each sample is a pool of cells from three different experimental animals per group. The two independent scRNA-seq data were integrated for the analysis. For the inset box plots, the horizontal line represents the median, the hinges denote the first and third quartiles, and the whiskers denote the minimum and maximum values. peTh2, pathogenic effector Th2; Th2 Trm pathogenic Th2 Tissue-resident memory; Treg, regulatory T cell.

described[47–50]. However, to fully dissect the biology of Treg/Th2 cells, lineage tracing experiments are warranted.

Collectively, our data resolve the heterogeneity of mouse lung pTh2 cells, reveal their similarities with lung *Il1rl1*-expressing Th cells (Treg/Th2), and highlight the need to define markers to distinguish the pTh2 cell subsets from the Treg/Th2 subset.

## Flow cytometric characterisation of lung pTh2 subsets in HDM-induced allergic asthma

Given some of the similarities we observed between the peTh2, Th2 Trm and Treg/Th2 subsets, particularly the expression of *Il1rl1* (Fig. 1g), it is important to define markers to distinguish them. This is crucial as ST2 is established as a marker to define pTh2 cells[15,18,42,51,52]. Our analyses suggest that using ST2 alone is insufficient to define pTh2 cells since lung Treg/Th2 cells also expressed the gene. To this end, we used the differentially expressed genes (DEGs) in peTh2, Th2 Trm and Treg/Th2 (Supplementary Data 2) and compared them to identify surface markers that are distinctively expressed by these subsets. Our comparison revealed a remarkable overlap between these three subsets (Fig. 2a and Supplementary Data 3). However, we found that Treg/Th2 cells were enriched for *Klrg1* and *Cd27* as compared with the pTh2 cell subsets (Fig. 2a, b). To validate our observation, we performed a flow cytometric analysis of lung CD4⁺ T cells from mice exposed to PBS or HDM. By using KLRG1, CD27 and additional surface markers (Fig. 2c and Supplementary Fig. 4), we first validated the expression of ST2 on KLRG1⁺ and CD27⁺ Th subsets (Treg/Th2 cells) (Fig. 3a–c). Unexpectedly, we found IL-13 expression in some of the ST2⁺KLRG1⁺ and ST2⁺CD27⁺ Th cells (Fig. 3b, c). These cells also expressed Foxp3 and Gata3 (Fig. 3d, e).

Based on the above observations (Fig. 3a–e), we sought to establish a flow cytometry gating strategy to properly define lung pTh2 cells. We first used CD27 and KLRG1 to exclude the Treg/Th2 subset, and surprisingly, found additional heterogeneity within this Th population. Our flow cytometric analysis revealed three distinct populations of Treg/Th2 cells: (1) ST2⁺CD27⁺KLRG1⁻, (2) ST2⁺CD27⁺KLRG1⁺and (3) ST2⁺CD27⁻KLRG1⁺ (Fig. 3f). These Treg/Th2 populations contained cells expressing FoxP3 and IL-13 (Fig. 3g). We collectively refer to these populations of Treg/Th2 cells as non-pTh2 ST2⁺ Th cells due to their high FoxP3 expression. By excluding these non-pTh2 ST2⁺ Th cells, the remaining population of ST2⁺CD27⁻ KLRG1⁻ cells was highly enriched in IL-13⁺ Th cells with only residual FoxP3 expression (Fig. 3g), indicating that the typical pTh2 cells are within the ST2⁺CD27⁻KLRG1⁻ population. Thus, using CD27 and KLRG1 as markers to exclude non-pTh2 ST2⁺ Th cells (Fig. 3f and Supplementary Fig. 4), and gating the remaining cells based on CD69 and PD-1 as additional Trm marker[40] (Fig. 3h), we revealed that lung peTh2 cells can be identified as ST2⁺CD27⁻KLRG1⁻CD69ˡᵒʷPD1ʰⁱᵍʰ (Fig. 3i), and Th2 Trm cells as ST2⁺CD27⁻KLRG1⁻CD69ʰⁱᵍʰPD1ʰⁱᵍʰ (Fig. 3k). We confirmed that these cells are indeed pTh2 cells due to their high IL-13 protein expression (Fig. 3j, l). Consistently, human allergen-specific pTh2 (Th2A) cells also exhibit a reduced level of CD27 (refs. 12,53–55). These findings support our strategy to use CD27, together with KLRG1, as key markers to exclude non-pTh2 ST2⁺ Th cells. Overall, we have validated the presence of distinct lung ST2⁺ Th subsets and defined surface markers to comprehensively distinguish them from the pTh2 cell subsets. Most importantly, we anticipate that using the surface markers we have proposed (Fig. 3m), lung pTh2 cell subsets can independently be isolated for further investigation.

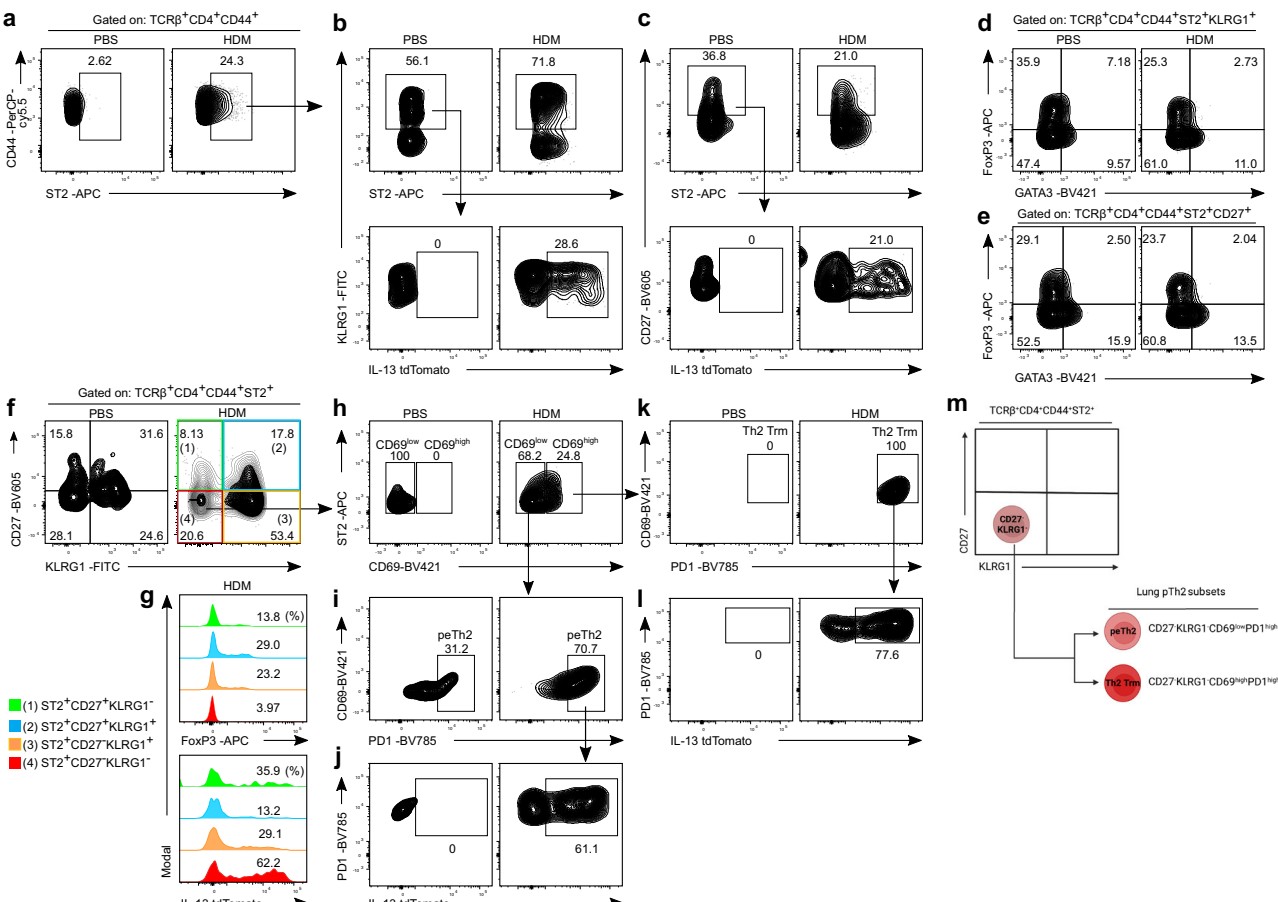

**Fig. 3 | Flow cytometric characterisation of lung pTh2 subsets in response to HDM. a–l** Flow cytometry characterisation of lung ST2+ Th cells in mice sensitised and challenged with PBS or HDM (as in Supplementary Fig. 1a). **a–c** Confirmation of ST2 expression by lung KLRG1+ and CD27+ Th cells. **a** Representative plots showing all lung ST2+ Th cells (gated on TCRβ+CD4+CD44+). **b** Representative plots showing ST2 against KLRG1 expression on Th cells gated on ST2 as depicted in (**a**) (top) and IL-13 expression by ST2+KLRG1+ Th cells (bottom). **c** Representative flow cytometry plots showing ST2 against CD27 expression on Th cells gated on ST2 as depicted in (**a**) (top) and IL-13 expression by ST2+CD27+ Th cells (bottom). **d** Representative plots of FoxP3 and GATA3 expression in lung Th cells (gated on TCRβ+CD4+CD44+ST2+KLRG1+). **e** Representative plots of FoxP3 and GATA3 expression in lung Th cells (gated on TCRβ+CD4+CD44+ST2+CD27+). **f, g** Defining the distinct subsets of lung ST2+ Th cells using CD27 and KLRG1 as markers. **f** Representative plots showing CD27 and KLRG1 expression by lung Th cells (gated on

TCRβ+CD4+CD44+ST2+). **g** Histograms showing the expression of FoxP3 (top) and IL-13 (bottom) by the different ST2+ Th subsets in mice exposed to HDM. **h–l** Gating strategy to define lung peTh2 and Th2 Trm cells. **h** Flow cytometry plots of CD69 expression (CD69low and CD69high) by CD27-KLRG1- Th cells in (**f**). **i** Flow cytometry plots showing PD1 expression by CD69low cells in (**h**), representing peTh2 cells. **j** IL-13 expression by the peTh2 cells in (**i**). **k** Flow cytometry plots showing PD1 expression by CD69high cells in (**h**), representing Th2 Trm cells. **l** IL-13 expression by the Th2 Trm cells in (**k**). **m** The schematic summarises the proposed surface markers to distinguish lung peTh2 cells (TCRβ+CD4+CD44+ST2+CD27-KLRG1-CD69lowPD1high) and Th2 Trm cells (TCRβ+CD4+CD44+ST2+CD27-KLRG1-CD69highPD1high) from non-pTh2 ST2+ Th cells (ST2+FoxP3+ Th cells: ST2+CD27+KLRG1-, ST2+CD27+KLRG1+, ST2+CD27-KLRG1+). Created in BioRender. Khan, M. (2025) https://BioRender.com/ w45h785. Data are representative of three independent experiments.

## Loss of HDAC1 augments the pathogenicity of pTh2 subsets and Th2 Trm cell generation

To determine the contribution of HDAC1 in the differentiation and effector function of lung pTh2 cell subsets, we grouped the cells by their origin (based on HTO labelling) (Fig. 4a, right). As expected, the majority (more than 80%) of cells in the two pTh2 cell clusters were IL-13+ Th cells from mice exposed to HDM (Fig. 4a, right, and Supplementary Data 1). Notably, despite the low expression of *Il13* mRNA in the Th2 Trm cell subset (Supplementary Fig. 3), these cells expressed IL-13 protein (Fig. 4b and Supplementary Data 1). Additionally, we found that about 35% of cells in the Treg/Th2 cluster composed of IL-13+ Th cells (Supplementary Data 1). This is in line with our flow cytometric analysis that some of the non-pTh2 ST2+ Th cells expressed IL-13 (Fig. 3a–e). These results further strengthen our strategy to exclude non-pTh2 ST2+ Th cells from the typical pTh2 cell subsets. Furthermore, we observed differences in the distribution of IL-13+ Th cells from WT and HDAC1-cKO mice exposed to HDM in the peTh2 (cluster 2) (Fig. 4b, top) and Th2 Trm

(cluster 5) (Fig. 4b, bottom) cell subsets. Most of the IL-13+ Th cells in the peTh2 cell subset were from WT mice, while the Th2 Trm cell subsets consisted of more IL-13+ Th cells from HDAC1-cKO mice (Fig. 4b and Supplementary Data 1). These observations suggest distinct differentiation stages between the two pTh2 cell subsets and a potential role for HDAC1 in regulating the processes involved. Further analysis of selected effector molecules and TFs showed a distinct expression profile of Th2 cytokines and other proinflammatory mediators between peTh2 and Th2 Trm cells, with many of the genes already downregulated in the Th2 Trm cells (Fig. 4c, d).

An important aspect of allergic asthma is the generation of long-lived allergen-specific Th2 Trm cells that rapidly respond upon allergen re-exposure to orchestrate type 2 immune responses in the lung[15,41,42,51,52,56,57]. CD4+ Trm cells are defined by upregulation of CD69, CD49a and CD11a, and downregulation of CCR7, CD62L, S1PR1 and Klf2[40,58,59]. CD103, which is expressed by some CD8+ Trm subsets, is rarely detected on lung CD4+ Trm cells including Th2 Trm cells[41,51,59].

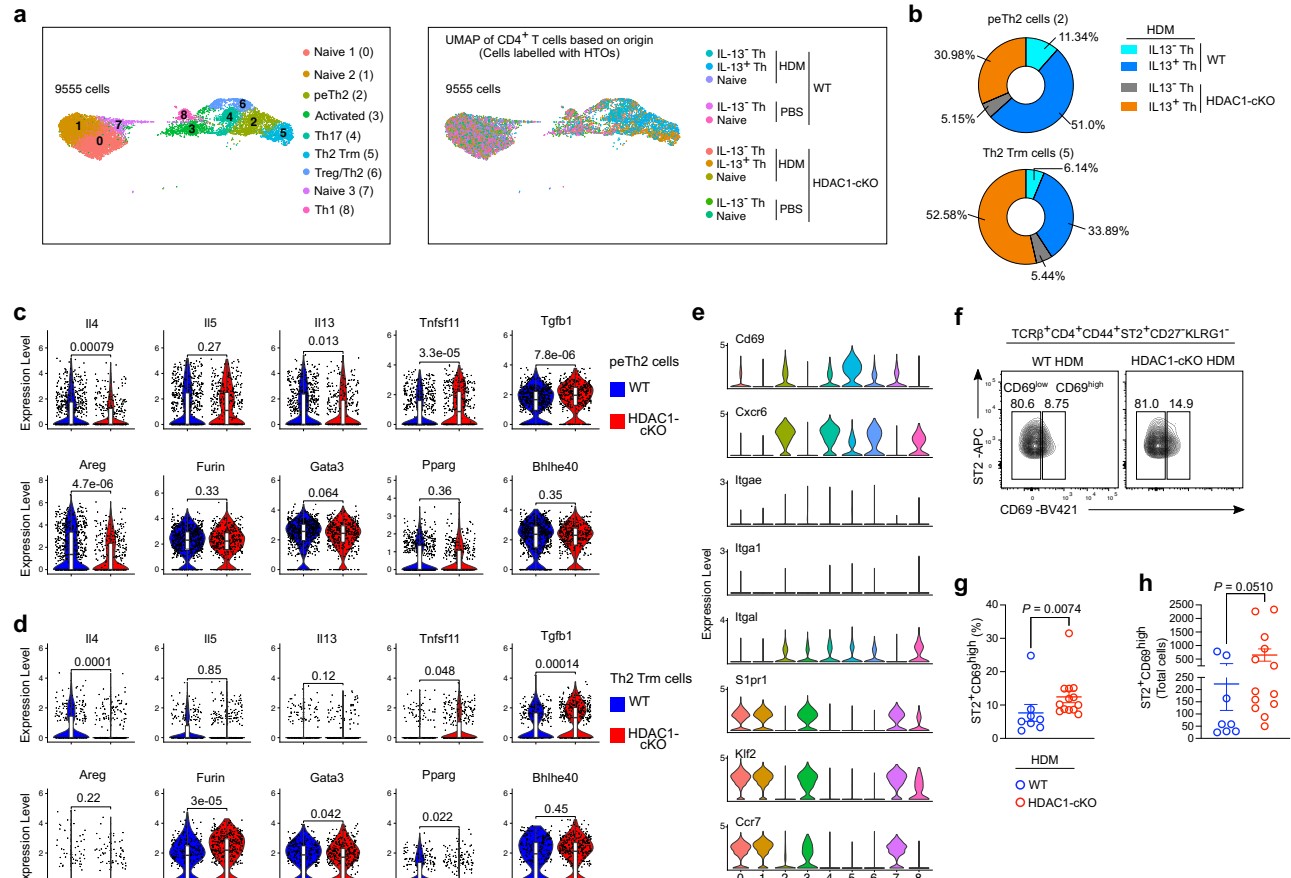

**Fig. 4 | Loss of HDAC1 augments the pathogenicity of pTh2 subsets and Th2 Trm cell generation. a** UMAP of lung CD4+ T cell clusters (left), and UMAP of lung CD4+ T cell coloured by origin (sorted samples labelled with unique HTOs), representing cells from HDAC1-cKO mice exposed to either HDM or PBS, and cells from WT mice exposed to either HDM or PBS (right). **b** Frequencies of IL-13⁻ Th and IL-13⁺ Th cells from WT and HDAC1-cKO mice exposed to HDM in the peTh2 (top) and Th2 Trm (bottom) clusters as in (**a**). **c, d** Violin plots of selected pathogenic marker genes in peTh2 cells (**c**) and Th2 Trm cells (**d**) from WT and HDAC1-cKO mice. **e** Violin plots of marker genes associated with tissue residency. **f** Representative flow cytometry plots of lung ST2+CD69^low and ST2+CD69^high cells

from WT and HDAC1-cKO mice sensitised and challenged to PBS or HDM. Cells in **f** are gated on TCRβ+CD4+CD44+ST2+CD27⁻KLRG1⁻. **g, h** Graphs show the frequency (**g**) and total number (**h**) of ST2+CD69^high Th cells in (**f**). For **f–h**, data are pooled from three independent experiments (n = 8 for WT; n = 13 for HDAC1-cKO). For **c** and **d**, two-tailed P-values were generated using the stat_compare_means (Wilcoxon test) function of the ggpubr package. For the inset box plots, the horizontal line represents the median, the hinges denote the first and third quartiles, and the whiskers denote the minimum and maximum values. For **g** and **h** data are presented as the mean ± SEM, and statistical analysis was performed using a two-tailed Mann-Whitney U test. Source data (**g, h**) are provided as a Source Data file.

Instead, Th2 Trm cells express ST2[15,42,51,52]. As expected, the Th2 Trm cell subset showed a signature of tissue residency, as evidenced by their diminished expression of *S1pr1*, *Ccr7* and *Klf2* and enhanced expression of *Cd69* (Fig. 4e). To validate our transcriptomic data that Th2 Trm cells are increased in absence of HDAC1 (Fig. 4b), we performed flow cytometric analysis of lung cells from WT and HDAC1-cKO mice exposed to HDM or PBS. We confirmed that, loss of HDAC1 resulted in increased lung Th2 Trm cells (Fig. 4f–h). Importantly, given the increased proportion of HDAC1-cKO cells in the Th2 Trm cell subset (Fig. 4b, bottom) and in the lungs of HDAC1-cKO mice exposed to HDM (Fig. 4f–h), it is conceivable that HDAC1 plays a critical role in regulating the molecular processes involved in the formation of Th2 Trm cells. Collectively, these data underline the importance of HDAC1 in restraining the pathogenicity of lung pTh2 cell subsets as well as in the formation of Th2 Trm cells.

## Lung pTh2 subsets exhibit shared and distinct transcriptional signatures

To better characterise the transcriptional signatures of lung pTh2 cell subsets, we compared each pTh2 cluster (peTh2 and Th2 Trm) with all other clusters. Our comparison revealed enrichment of transcriptional regulators such as *Bhlhe40*, *Gadd45b*, *Gata3*, *Pparg*, *Rbpj*, *Socs2*, *Hlf*, *Rgs1*, *Nfat5* and *Zeb2* in both pTh2 cell subsets (Fig. 5a, b and

Supplementary Data 4). Previous studies have demonstrated important roles for *Pparg*, *Bhlhe40* and *Nfat5* in regulating type 2 cytokine production[16,17,60–63]. It will be crucial to further investigate the role of these transcriptional regulators in pTh2 cell differentiation particularly in Th2 Trm formation and maintenance.

Next, we compared the peTh2 cell subset with the Th2 Trm cell subset to define subset-specific gene signature. Interestingly, despite their overlapping transcriptional signatures, they differ in the expression of specific genes. The peTh2 cell subset was more enriched for effector molecules such as *Il4*, *Il5*, *Il10*, *Il13*, *Tnfsf11 (Rankl)*, *Areg* and *Calca*, as well as several members of the tumor necrosis factor receptor superfamily (TNFRSF) *Tnfrsf4* (OX40), *Tnfrsf9* (4-1BB) and *Tnfrsf18* (GITR) (Fig. 5c and Supplementary Data 5). In contrast to the peTh2 cell subset, the Th2 Trm cell subset showed diminished expression of pathogenic effector molecules, and an enrichment of *Lpar6* and *Slc38a2*, which are known to modulate lysophosphatidic acid and amino acid metabolism[64–66], as well as *Zfp36l2* (Fig. 5c and Supplementary Data 5). The Th2 Trm cell also showed increased expression of transcriptional regulators of the activator protein-1 (AP-1) family *Fos*, *Fosb* and *Jun* (Fig. 5c, d).

We further performed GSEA to define the pathways enriched in the pTh2 cell subsets compared with other CD4+ T cells. Our analysis

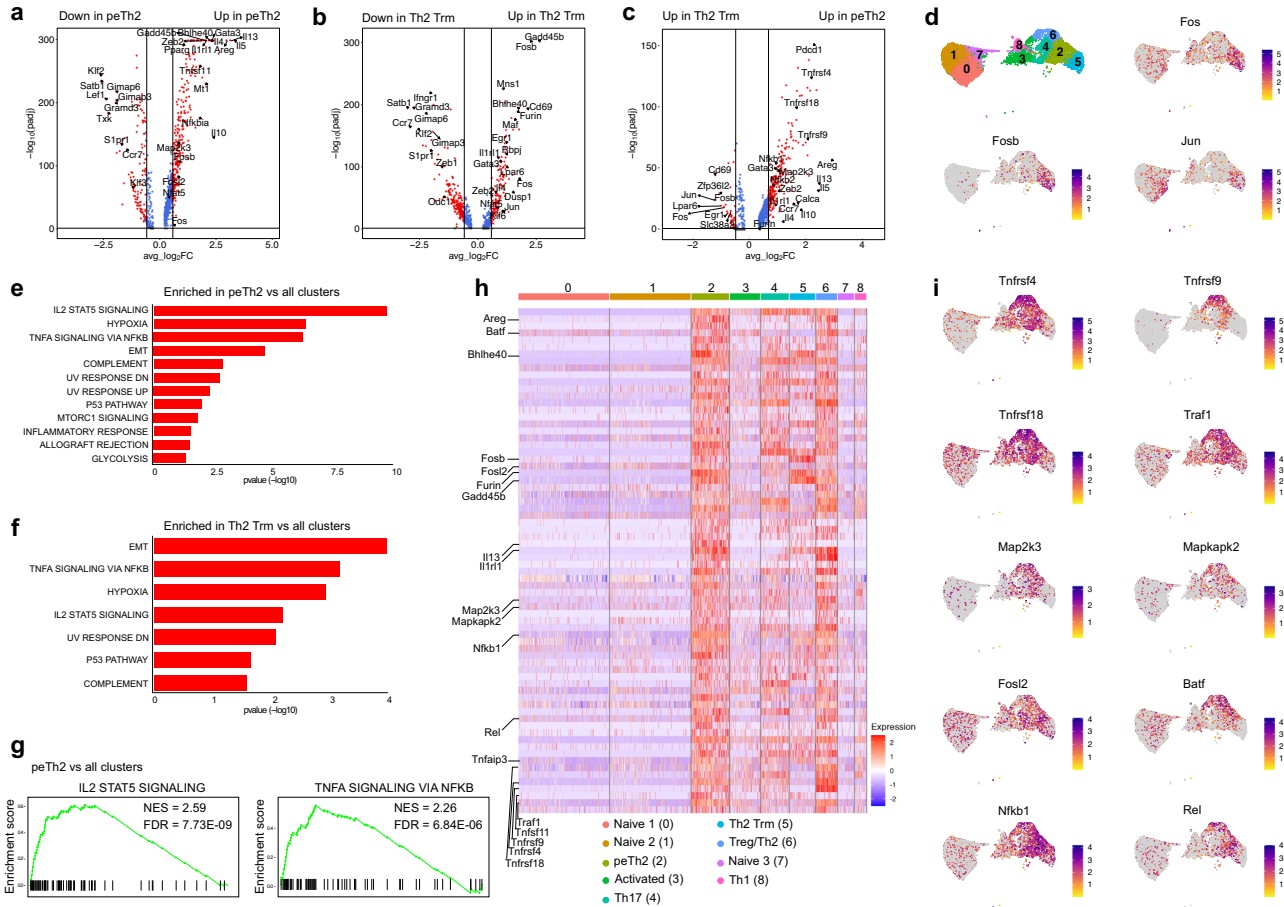

**Fig. 5 | Lung pTh2 subsets exhibit shared and distinct transcriptional signatures.** Volcano plots showing a comparison between peTh2 cells and all other clusters (**a**), Th2 Trm cells and all other clusters (**b**), and peTh2 cells and Th2 Trm cells (**c**). The vertical and horizontal lines indicate average Log2 Fold change of ≤ −0.58 and ≥ 0.58, and adjusted *P*-value < 0.05, respectively. Adjusted *P*-values were calculated using Seurat's default two-tailed Wilcoxon rank sum test with Bonferroni correction. **d** UMAP plots of CD4⁺ T cells colour-coded based on the log-normalised expression levels of selected genes. **e** GSEA of HALLMARK pathways upregulated (unadjusted *P*-value < 0.05) in peTh2 cells as compared with all other clusters. **f** GSEA of HALLMARK pathways upregulated (unadjusted *P*-value < 0.05) in

Th2 Trm cells as compared with all other clusters. **g** Enrichment plots of the IL-2/STAT5 (left) and TNF/NFκB (right) pathways in peTh2 cells (Fig. 5e). For (**e**–**g**), *P*-values were calculated using fgsea's adaptive multilevel splitting Monte Carlo scheme with Benjamini-Hochberg correction. **h** Heatmap shows the leading-edge genes for the pathways in (**g**). **i** UMAP plots of CD4⁺ T cells colour-coded based on the log-normalised expression levels of selected genes. IL-2 Interleukin-2, STAT5 signal transducer and activator of transcription factor 5, TNF tumour necrosis factor alpha, NFKB nuclear factor kappa B, EMT epithelial-mesenchymal transition, UV ultraviolet, FDR false discovery rate, NES normalised enrichment score.

revealed an enrichment of similar pathways in both pTh2 cell subsets (Fig. 5e, f and Supplementary Data 6). Notably, the IL-2/STAT5 and TNF/NF-κB signalling pathways were among the most upregulated. Further assessment of the IL-2/STAT5 and TNF/NF-κB leading-edge genes (the core genes that contribute to the enrichment of these pathways) in the peTh2 cells (Fig. 5g, h and Supplementary Data 6), revealed enhanced expression of genes associated with TNF (*Tnfrsf4*, *Tnfrsf9*, *Tnfrsf18 and Traf1*), mitogen-activated protein kinase (MAPK; *Map2k3*, Mapkapk2), AP-1 (*Fosl2*, *Batf*), and nuclear factor-κB (NF-κB; *Nfkb1*, *Rel*) signalling (Fig. 5h, i). These signalling components are known to regulate various cellular processes such as survival, stress response, proliferation and differentiation[67,68], but their role in pTh2 cell differentiation and effector function is unclear. Together, our analyses unveil the transcriptional profiles of lung peTh2 and Th2 Trm cells and identify mediators associated with their differentiation and effector function.

## Co-stimulation of GITR and TSLPR drives in vitro differentiation of pTh2 cells

A major limitation to our understanding of how pTh2 cells are regulated is the lack of an appropriate system to generate and investigate

them in vitro. We observed that lung peTh2 cells were highly enriched for *Tnfrsf4*, *Tnfrsf9* and *Tnfrsf18*, members of the TNFRSF[69]. Specifically, the activation of TNFRSF18 also called glucocorticoid-induced TNFR family-related protein (GITR), has been shown to promote the features of allergic asthma and the differentiation of Th2 cells in vitro[70–72]. In addition, we found upregulation of the IL-2/STAT5 pathway in both pTh2 cell subsets. Previous studies have shown that the epithelial-derived cytokine, thymic stromal lymphopoietin (TSLP), promotes pathogenic features in in vitro differentiated Th2 cells via the activation of STAT5 (refs. 73,74). Therefore, our data and previous work suggest that co-stimulation of GITR and/or TSLP receptor (TSLPR) could induce the in vitro generation of pTh2 cells. To test our hypothesis, we differentiated naïve CD4⁺ T cells under Th2-promoting conditions alone (Fig. 6a, left), or in the presence of TSLP alone, a GITR agonist (DTA-1) alone, or in combination (Fig. 6a, right). Our flow cytometric analyses revealed that both the GITR agonist and TSLP induced IL-4 and IL-13 expression in the Th2 cells, with the GITR agonist being more potent in inducing IL-13 expression (Fig. 6b, c). However, a combination of both stimuli resulted in higher expression of IL-4 and IL-13 (Fig. 6b, c). Remarkably, the level of IL-5 was more pronounced when both the GITR agonist and TSLP were combined

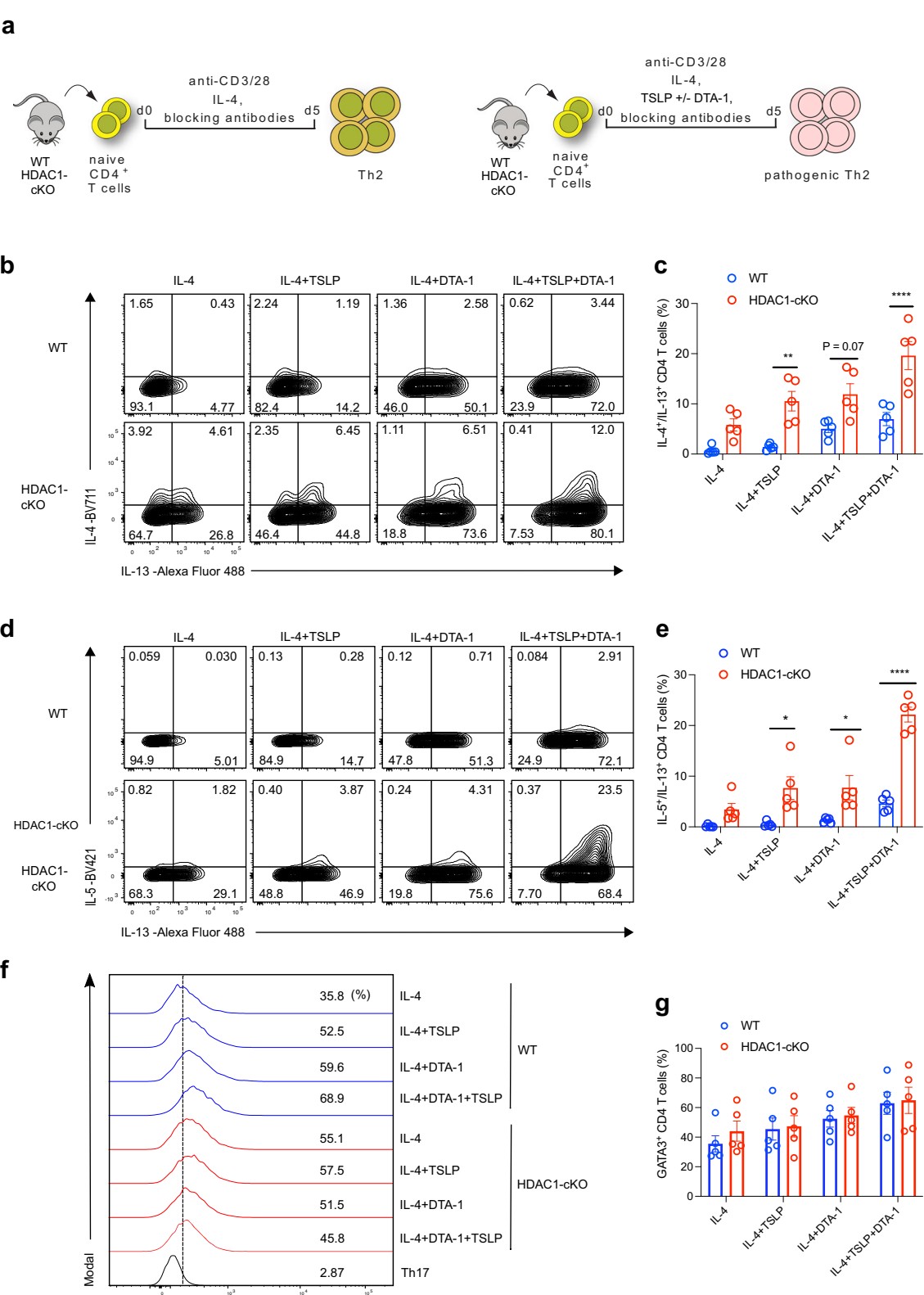

(Fig. 6d, e). Importantly, the deletion of HDAC1 resulted in an augmented expression of the pathogenic Th2 cytokines compared with WT cells (Fig. 6b–e) despite comparable GATA3 protein levels (Fig. 6f, g). The in vitro generated pTh2 cells also showed enhanced expression of RANKL, GM-CSF and IL-9 (Supplementary Fig. 5a–c). These results indicate that co-stimulation of GITR and TSLPR promotes in vitro differentiation of pTh2 cells.

We also assessed the impact of TNFRSF4 (OX40) and TNFRSF9 (4-1BB) co-stimulation on Th2 cell differentiation. Using anti-mouse OX40 and 4-1BB agonistic antibodies together with TSLP, we found that OX40 co-stimulation had a minimal effect in inducing pathogenic features in the in vitro differentiated Th2 cells (Supplementary Fig. 5d, e). Unlike OX40, 4-1BB co-stimulation led to enhanced expression of pathogenic cytokines (Supplementary Fig. 5d, e).

**Fig. 6 | Co-stimulation of GITR and TSLPR drives in vitro differentiation of pTh2 cells. a** Schematic of in vitro differentiation of Th2 cells (left) and pTh2 cells (right). Naïve CD4$^+$ T cells (TCRβ$^+$CD4$^+$CD62L$^+$CD44$^-$) from WT and HDAC1-cKO mice were activated with anti-CD3 and anti-CD28 in the presence of Th2-promoting conditions (IL-4, IL-2, anti-IFN-γ, and anti-TGF-β; collectively referred as IL-4), or Th2-promoting conditions plus TSLP alone, anti-GITR antibody (DTA-1) alone, or both TSLP and DTA-1, then cultured for 5 days. On day 5, cells were restimulated with PMA and ionomycin in the presence of GolgiStop and GolgiPlug for 4 h, and cytokine analyses were performed by flow cytometry. **b–g** Flow cytometric analysis of cells differentiated under conditions indicated in (**a**). **b** Representative plots showing the expression of IL-4 and IL-13 in WT cells (top) and HDAC1-cKO cells (bottom). **c** Graph shows the frequency of IL-4 and IL-13 co-expressing cells in (**b**). **d** Representative plots showing the expression of IL-5 and IL-13 in WT cells (top) and

HDAC1-cKO cells (bottom). **e** Graph shows the frequency of IL-5 and IL-13 co-expressing cells in (**d**). **f** Histograms depicting the expression of GATA3 in WT cells (blue) and HDAC1-cKO cells (red). **g** Graph shows the frequency of GATA3 expressing cells in (**f**). For **c** and **e**, the exact *P*-values are (IL-4 + TSLP = 0.0059; IL-4 + DTA-1 = 0.0707; IL-4 + TSLP + DTA-1 = < 0.0001) and (IL-4 + TSLP = 0.0151; IL-4 + DTA-1 = 0.0382; IL-4 + TSLP + DTA-1 = <0.0001), respectively. Th17 cells were used as controls for gating. Data are pooled from five independent experiments and presented as the mean ± SEM. Each symbol represents one mouse. Statistical analysis was performed using a Two-way ANOVA with Tukey's multiple comparisons test. *$P < 0.05$, **$P < 0.01$, ****$P < 0.0001$. The schematics in (**a**) were created using Illustrator. TSLP, thymic stromal lymphopoietin; GITR, glucocorticoid-induced TNFR-related protein. Source data (**c**, **e**, **g**) are provided as a Source Data file.

Although 4-1BB co-stimulation promoted the expression of pathogenic cytokines, it was less potent than GITR co-stimulation (Supplementary Fig. 5d, e). Collectively, these results reveal the crucial role of the TNFRSF members GITR and 4-1BB in driving Th2 cells toward a pathogenic state. Notably, we demonstrate that co-stimulation of GITR and TSLPR promotes the acquisition of strong pathogenic features in Th2 cells and represents an additional strategy to generate pTh2 in vitro. Our results further demonstrate that HDAC1 is essential to restrict the acquisition of pathogenic features, particularly IL-5, in cells differentiated under pTh2-promoting conditions.

## Combined transcriptome and proteome profiling reveals a shared signature between in vitro generated pTh2 and lung peTh2 cells

Next, we performed bulk RNA-seq to better define the transcriptional profile of pTh2 cells generated in vitro and to compare them with lung pTh2 cell subsets (Fig. 7a). Differential gene expression analysis of in vitro WT pTh2 and WT Th2 cells revealed upregulation of key pathogenic signature genes including *Bhlhe40*, *Pparg*, *Gadd45b*, *Zeb2*, *Atf3*, *Areg*, *Csf2*, *Il4*, *Il5*, *Il9* and *Il13* in pTh2 cells (Fig. 7b and Supplementary Data 7). A similar transcriptional signature was observed in HDAC1-cKO pTh2 cells compared with HDAC1-cKO Th2 cells (Fig. 7c and Supplementary Data 7). The absence of HDAC1 augmented the expression of pathogenic molecules (*Il5*, *Il13 and Bhlhe40*) in Th2 and pTh2 cells compared with WT Th2 and pTh2 cells, respectively (Supplementary Fig. 6a, b).

To determine whether the in vitro pTh2 cells exhibit a similar transcriptional signature as lung pTh2 cells, we first compared the lung peTh2 gene set (Supplementary Data 2) with gene sets from in vitro pTh2 and Th2 cells (Supplementary Data 7). Our GSEA revealed that both WT and HDAC1-cKO in vitro pTh2 cells were highly enriched for lung peTh2 signature as compared with their corresponding Th2 cells (Fig. 7d, e and Supplementary Data 8). Next, we took the leading-edge genes (the core genes that contribute to the enrichment of peTh2 signature) in WT pTh2 cells (Fig. 7d) and examined their expression profile across the in vitro Th2 and pTh2 cells from WT and HDAC1-cKO. Our analysis revealed that these leading-edge genes are distinctively expressed by pTh2 cells as compared with Th2 cells (Supplementary Fig. 6c and Supplementary Data 9). Thus, these genes represent the core pathogenic Th2 signature genes since they are highly enriched in lung peTh2 cells and in vitro pTh2 cells. Notably, some of the core pathogenic Th2 signature genes including *Il5*, *Il13*, *Bhlhe40*, *Map2k3* and *Mapkapk2* were also upregulated in HDAC1-cKO Th2 cells (Supplementary Fig. 6c), indicating that these cells have already acquired a pathogenic signature. Consistently, the HDAC1-cKO Th2 were also enriched for genes from the lung peTh2 gene set as compared with the WT Th2 cells (Fig. 7f and Supplementary Data 8), demonstrating that the absence of HDAC1 predisposes the in vitro Th2 cells to acquire a pathogenic program. These results further underline the crucial role of HDAC1 in restraining the differentiation of pTh2 cells. We further compared the in vitro pTh2 cells with lung Th2 Trm

cells. Although, the in vitro pTh2 cells shared some features with lung Th2 Trm cells, their similarity was less pronounced (Supplementary Fig. 6d–f and Supplementary Data 8) as compared with the lung peTh2 cells (Fig. 7d–f). This is expected as the Th2 Trm cells already acquired tissue residency features with diminished effector program (Fig. 4d, e). Overall, these results indicate that the pTh2 cells we have generated in vitro are highly comparable to lung peTh2 cells. Of note, the in vitro generated pTh2 cells do not express *Il1rl1* (ST2), indicating that our culture conditions are insufficient to induce ST2. Thus, additional signal (s), or longer time in culture might be required for ST2 expression as previously demonstrated[75]. Nonetheless, the in vitro pTh2 cells also showed an enrichment of similar pathways as the lung pTh2 cell subsets, notably, the IL-2/STAT5 and TNF/NF-κB pathways (Supplementary Fig. 6g, h and supplementary Data 10).

Our transcriptomic analyses revealed that lung and in vitro generated pTh2 cells are highly enriched in components of the AP-1, MAPK, and NF-κB pathways (Fig. 5h, i and Supplementary Fig. 6c). To validate these findings on protein level, we performed proteomic analysis by quantitative mass spectrometry using the in vitro generated pTh2 cells (Fig. 7g). Consistent with our transcriptomic findings, our proteomic analysis revealed enrichment of proteins associated with AP-1, MAPK, and NF-κB signalling in the in vitro pTh2 cells (Fig. 7h–j and supplementary Data 11). Overall, these findings reveal that lung peTh2 cells and pTh2 cells generated in vitro have a similar transcriptional signature and rely on related gene regulatory networks for their differentiation.

## The p38 MAPK pathway regulates IL-5 and IL-13 expression in pTh2 cells

Given that both lung and in vitro pTh2 cells showed upregulation of AP-1 and MAPK components, we assessed the impact of these pathways on pTh2 cell differentiation using small molecule inhibitors targeting AP-1 (T-5224) and p38 MAPK (SB 203580). SB 203580 markedly suppressed IL-5 and IL-13 expression in WT pTh2 without affecting IL-4 and GATA3 levels (Fig. 8a–c), and similar results were obtained using HDAC1-cKO pTh2 cells (Fig. 8d–f). Furthermore, SB 203580 suppressed the expression of GM-CSF and IL-9 in WT pTh2 cells (Supplementary Fig. 7a, b). In contrast, the inhibition of AP-1 with T-5224 did not affect WT and HDAC1-cKO pTh2 cell differentiation (Fig. 8a–f and Supplementary Fig. 7a, b). Moreover, targeting ERK1/2 and JNK signalling using the inhibitors U0126-EtoH and SP600125, respectively, did not affect pTh2 cell differentiation (Supplementary Fig. 7c, d). These results demonstrate that the p38 MAPK pathway selectively regulates IL-5 and IL-13 expression in pTh2 cells to restrict their differentiation and function.

## Regulation of type 2 cytokines in pathogenic Th2 cells at the chromatin level

An intriguing observation was the transcriptional upregulation and subsequent increased protein production of pathogenic type 2 cytokines in WT and HDAC1-cKO pTh2 cells as well as in HDAC1-cKO Th2

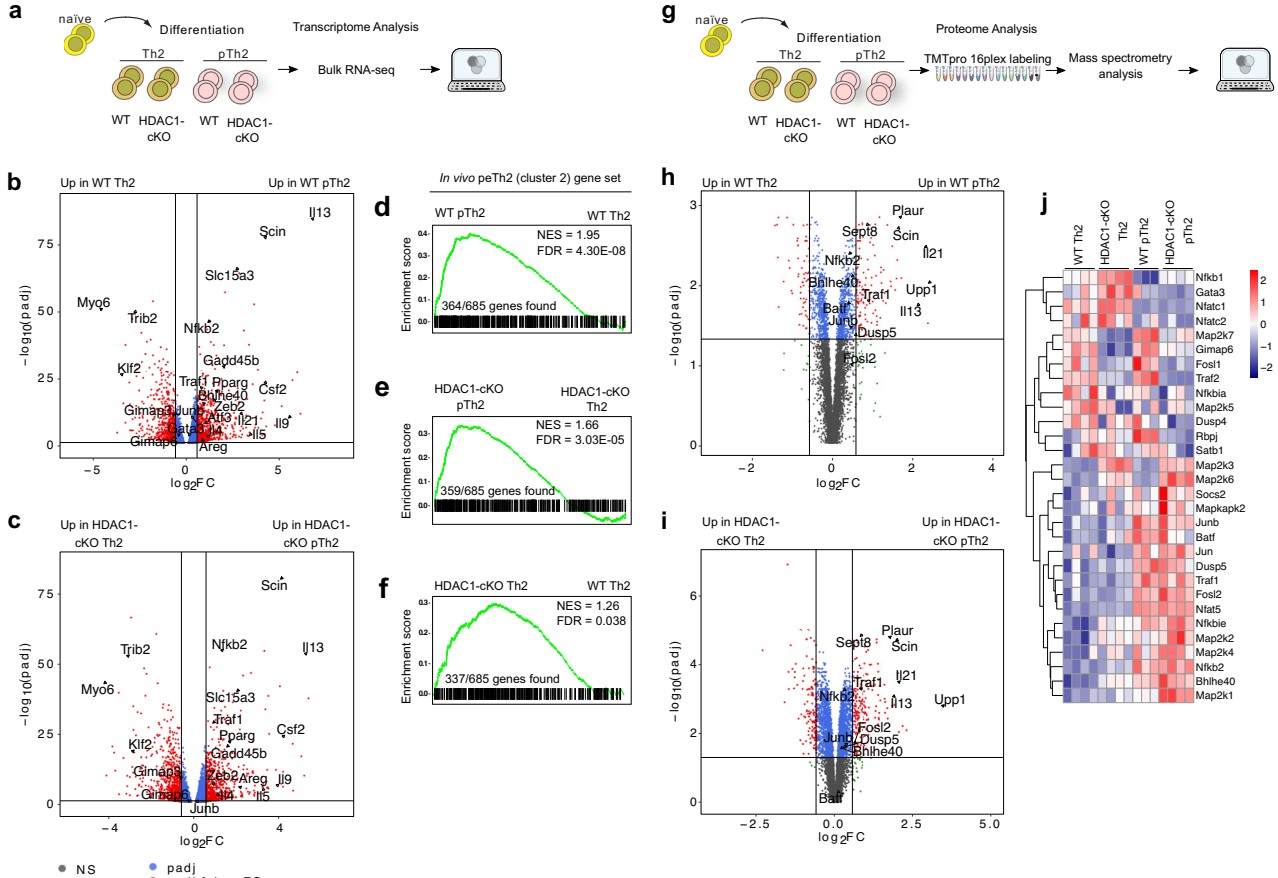

**Fig. 7 | Combined transcriptome and proteome profiling reveals a shared signature between in vitro generated pTh2 and lung peTh2 cells. a** Schematic of Bulk RNA-seq of Th2 and pTh2 cells from WT and HDAC1-cKO after 72 h in culture. **b** Volcano plot of DEGs (adjusted *P*-value < 0.1) between WT pTh2 and WT Th2 cells. **c** Volcano plot of DEGs between HDAC1-cKO pTh2 cells and HDAC1-cKO Th2 cells (adjusted *P*-value < 0.1). For (**b**, **c**), two-tailed *P*-values are based on DESeq2's Wald test and adjusted using the Bioconductor Independent Hypothesis Weighting package. **d**–**f** GSEA of in vitro generated pTh2 cells to lung peTh2 cells. **d** Enrichment plot showing a comparison of WT pTh2 cells and WT Th2 cells to lung peTh2 cells. **e** Enrichment plot showing comparison of HDAC1-cKO pTh2 cells and HDAC1-cKO Th2 cells to lung peTh2 cells. **f** Enrichment plot showing comparison of HDAC1-cKO Th2 cells and WT Th2 cells to lung peTh2 cells. DEGs (adjusted *P*-value < 0.1; two-tailed *P*-values obtained by DESeq2's Wald test and adjusted using the Bioconductor Independent Hypothesis Weighting package) between WT pTh2 vs WT Th2, HDAC1-cKO pTh2 vs HDAC1-cKO Th2, and HDAC1-cKO Th2 vs WT Th2

cells (Supplementary Data 7) were used to compare with lung peTh2 gene set (DEGs; adjusted *P*-value < 0.05 based on Seurat's two-tailed Wilcoxon rank sum test with Bonferroni correction; Supplementary Data 2). For the enrichment plots in (**d**–**f**), *P*-values were calculated using fgsea's adaptive multilevel splitting Monte Carlo scheme with Benjamini-Hochberg correction. **g** Schematic of proteomics analysis of in vitro generated Th2 and pTh2 cells from WT and HDAC1-cKO as in (**a**). Volcano plots depicting proteomics comparison between WT pTh2 and WT Th2 cells (**h**) and HDAC1-cKO pTh2 and HDAC1-cKO Th2 cells (**i**). For (**h**, **i**) two-tailed *P*-values were obtained by Limma moderated t-test with Benjamini-Hochberg correction. **j** Heatmap showing normalised reporter ion intensity values of selected proteins. For (**b**, **c**, **h**, **i**) the vertical and horizontal lines indicate Log2 Fold change of ≤−0.58 and ≥0.58, and adjusted *P*-value < 0.05, respectively. Transcriptomic and proteomic data are from three and four independent experiments, respectively. One WT pTh2 sample was excluded from the proteomics analyses due to poor sample quality.

cells but not in WT Th2 cells, and without a significant change in GATA3 expression between all the groups analysed. This might suggest that under pTh2 differentiation conditions, and in absence of HDAC1, chromatin accessibility is changed at the Th2 cytokine locus, thereby allowing a higher recruitment of GATA3 and additional transcriptional regulators that are crucial for cytokine transcription. To test this hypothesis, we leveraged our in vitro culture system to perform the Assay for Transposase-Accessible Chromatin using sequencing (ATAC-seq)[76] on WT and HDAC1-cKO Th2 and pTh2 cells after 24 and 48 h of culture. Comparison of type 2 cytokine gene loci revealed a strong increase in chromatin accessibility at 48 h of culture (Fig. 9a), thus, we focused our analysis on this time point.

UMAP analysis showed sample clustering based on genotype and differentiation condition after 48 h, with pTh2 cells from WT and HDAC1-cKO clustering together as compared to WT Th2 cells, while the HDAC1-cKO Th2 cells exhibited a bias towards a pathogenic state (Fig. 9b). This suggests that the induction of a pathogenic program in Th2 cells

involves the inactivation or overriding of HDAC1 function, resulting in increased chromatin accessibility at the Th2 cytokine locus. Thus, we specifically compared accessibility of Th2-specific loci in proximity of genes coding for *Il4, Il13, Rad50* and *Il5* in WT Th2 to the other three conditions (three conditions defined as pathogenic or transitioning to a pathogenic state due to the loss of HDAC1) (Fig. 9c). Six peaks were identified as significantly higher in the pathogenic/transitioning pathogenic state compared to the non-pathogenic one (WT Th2).

To identify potential TFs that might regulate peTh2 cell differentiation in vivo, we utilised our scRNA-seq data and compared the peTh2 cells (cluster 2) versus all other effector Th cell clusters (clusters 3,4,5,6 and 8) (Fig. 9d). Our comparison revealed an enrichment of the type 2 cytokines *Il4, Il13* and *Il5* as well as *Gata3* in the peTh2 cells, which are hallmarks of fully differentiated effector Th2 cells[77]. In addition, several other TFs including *Bhlhe40, Nfkb1, Rel, Rbpj, Zeb2, Pparg, Fosl2, Junb* and *Atf3* were significantly upregulated in these cells. Using the list of differentially enriched TFs from (Fig. 9d), we performed a TF binding

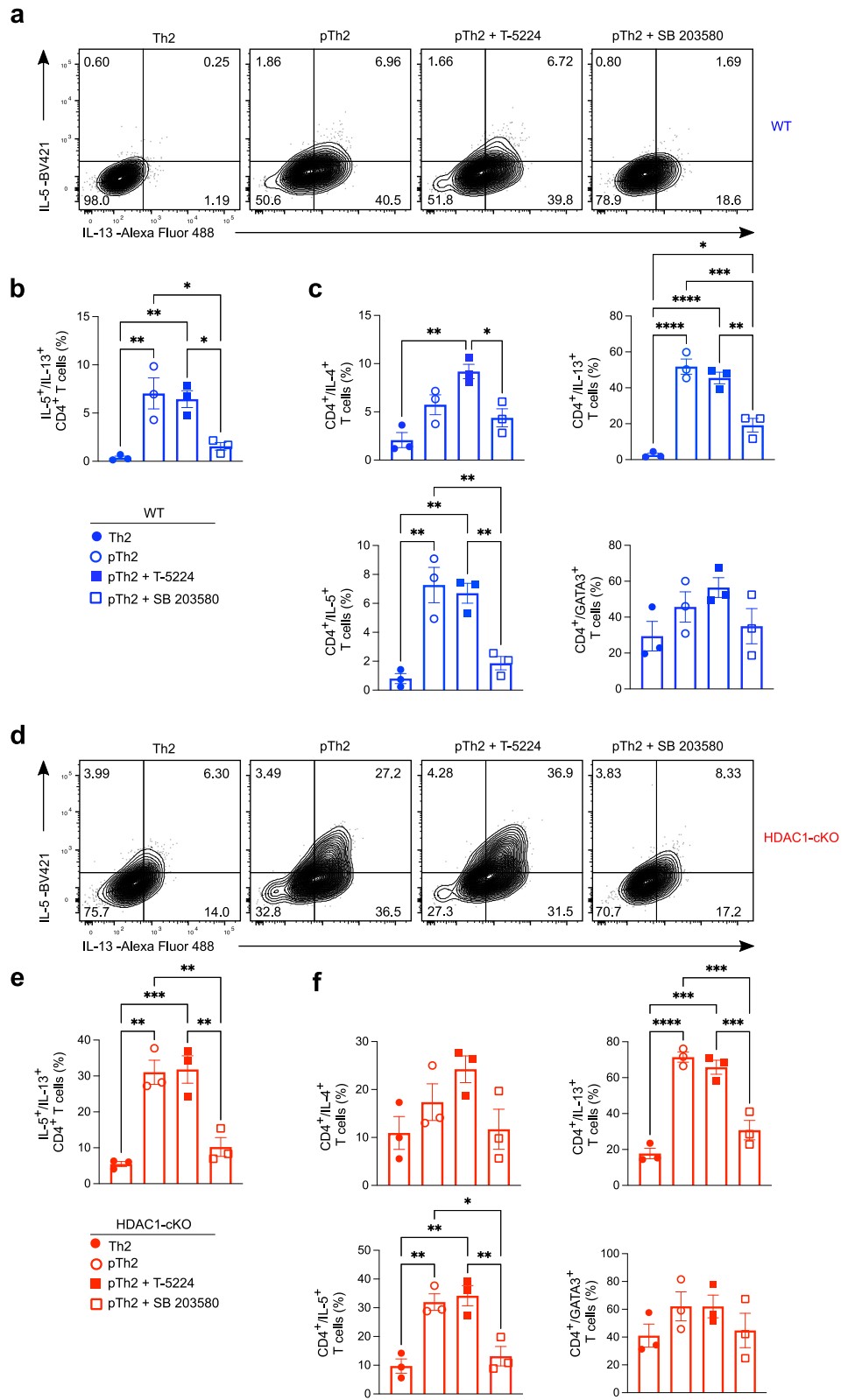

motif analysis using chromVAR on chromosome 11 as genome-wide representative[78]. Importantly, we observed an enhanced binding potential for GATA3, NFκB1, REL, RBPJ, ATF3, PPARγ Fosl2, JUNB and JUND in pTh2 compared to Th2 (Fig. 9e). Thus, we found a strong correlation between expression and binding potential to open chromatin sites. However, further experiments are warranted to confirm whether these TFs specifically bind to the Th2 cytokine locus.

Collectively, our data suggest a model (Fig. 9f) whereby the establishment of a pathogenic state in Th2 cells is accompanied by an increased accessibility of the Th2 cytokine locus enabling the binding of GATA3 and additional pTh2-driving TFs like PPARγ[16,17,60]. Our finding that HDAC1-cKO classical Th2 cells exhibit enhanced opening of Th2 cytokine gene loci similar to WT and HDAC1-cKO pTh2 cells, indicates that HDAC1 is restraining the accessibility to type 2 cytokine gene loci

**Fig. 8 | The p38 MAPK pathway regulates IL-5 and IL-13 expression in pTh2 cells.** **a**–**f** Impact of targeting AP-1 or p38 MAPK pathways on pTh2 cells generated in vitro. Th2 cells alone, pTh2 cells alone, pTh2 cells treated with an AP-1 inhibitor (T-5224; 10 μM), and pTh2 cells treated with a p38 MAPK inhibitor (SB 203580; 10 μM) were cultured for five days under Th2 and pTh2-promoting conditions (Fig. 7a). On day 5, cells were restimulated with PMA and ionomycin in the presence of GolgiStop and GolgiPlug for 4 h before cytokine analyses by flow cytometry. Flow cytometric analysis of WT cells. **a** Representative flow cytometry plots showing IL-5 and IL-13 expression under indicated conditions. **b** Graph shows the frequency of IL-5 and IL-13 co-expressing cells in (**a**). **c** Graphs showing the

frequencies of IL-4, IL-5, IL-13, and GATA3 single-expressing cells. **d**–**f**, Flow cytometric analysis of HDAC1-cKO cells. **d** Representative flow cytometry plots showing IL-5 and IL-13 co-expression. **e** Graph shows the frequency of IL-5 and IL-13 co-expressing cells in (**d**). **f** Graphs showing frequencies of IL-4, IL-5, IL-13, and GATA3 single-expressing cells. Data are pooled from three independent experiments and presented as the mean ± SEM. Each symbol represents one mouse. Statistical analysis was performed using a one-way ANOVA with Tukey's multiple comparisons test. $^{*}P < 0.05$, $^{**}P < 0.01$, $^{***}P < 0.001$, $^{****}P < 0.0001$. AP-1, activator protein-1; p38 MAPK, p38 mitogen-activated protein kinase. Source data (**b**, **c**, **e**, **f**) are provided as a Source Data file.

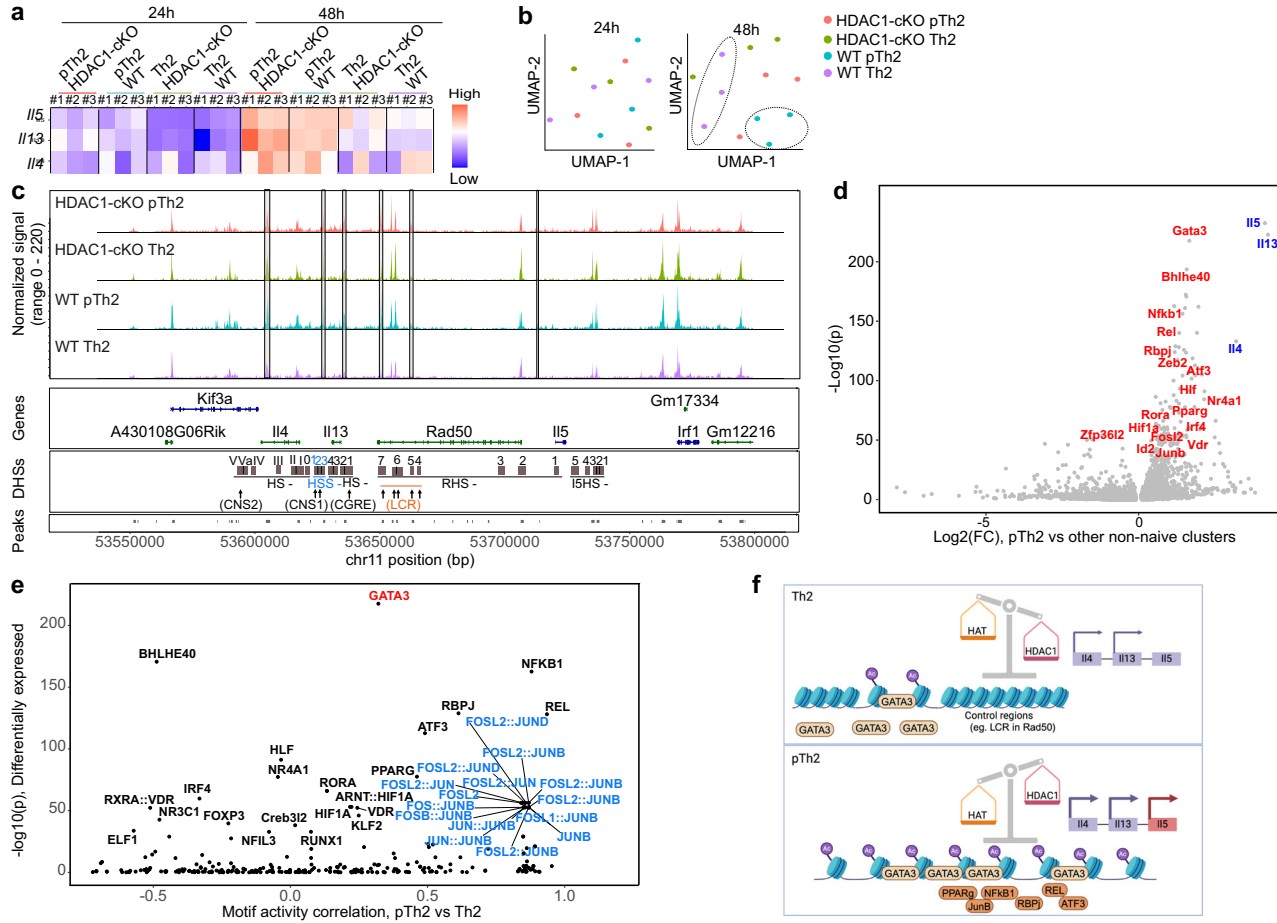

**Fig. 9 | Type 2 cytokine expression is differentially regulated at chromatin level in pathogenic versus classical Th2 cells. a** Heatmap representing normalised ATAC-seq peaks of Th2 cytokines from WT and HDAC1-cKO Th2 and pTh2 cells generated in vitro after 24 or 48 h of culture. **b** UMAP of ATAC-seq profiles of WT and HDAC1-cKO Th2 and pTh2 cells that were cultured for 24 (left panel) or 48 h (right panel). Clustering of WT Th2 and pTh2 are highlighted with dashed circles at 48 h. **c** ATAC-seq signal tracks at the Th2 cytokine locus in WT and HDAC1-cKO Th2 and pTh2 cells generated in vitro after 48 h of culture. The average signals from three replicates are shown as one track. Statistically significant differences are highlighted as grey bars. Relative positions of the integration sites to the known DNase I hypersensitive sites (squares) and conserved noncoding regions/elements/control regions (arrows) are shown. HSS and LCR were coloured for a better

visualisation. **d** Volcano plot showing a comparison between peTh2 cells (cluster 2) from scRNA-seq analysis versus all other effector ("non-naïve") clusters. Major Th2 related Transcription factors are shown in red and cytokines in blue. **e** Volcano plot of chromVAR inferred TF binding sites between pTh2 and Th2 cells generated in vitro. GATA3 and a prominent AP-1 TF node were coloured to improve visualisation. **f** Schematic representation of the working model. Created in BioRender. Boucheron, N. (2025) https://BioRender.com/x77f144. In summary, conventional Th2 cells have a reduced chromatin accessibility, with high HDAC1 activity and low *Il4* and *Il13* expression. In contrast, pTh2 cells have enhanced chromatin accessibility, with a reduced/overridden HDAC1 activity coupled with high *Il4*, *Il13*, and *Il5* expression.

for GATA3 and additional TFs, and that under pathogenic conditions HDAC1 is at least partially inactivated or overridden at the Th2 cytokine locus.

## Discussion

Unlike classical Th2 cells, pTh2 cells secrete high amounts of type 2 cytokines. They have become an important drug target in allergic asthma. Thus, defining the factors orchestrating their differentiation

and pathogenicity is of great clinical importance. Our data resolve the heterogeneity of mouse pTh2 cells, their transcriptional signatures, and uncover similarities between these cells and other lung ST2⁺ Th cells. Additionally, we establish a protocol to generate pTh2 cells in vitro, allowing the investigation of pathways and regulators critical for their differentiation. Furthermore, we find that HDAC1 is essential to restrict HDM-induced allergic asthma and the differentiation of lung and in vitro generated pTh2 cells (Supplementary Fig. 8). Our study

further indicates that HDAC1 restricts chromatin accessibility at the Th2 cytokine locus.

Our scRNA-seq analysis reveals two distinct subsets of lung pTh2 cells: peTh2 and Th2 Trm cells. Both subsets are highly proinflammatory and express various inflammatory cytokines and mediators such as *Il4*, *Il5*, *Il13*, *Tnfsf11*, *Areg*, *Tgfb1*, *Calca* and *Furin*, all of which are implicated in promoting allergic asthma and other allergic diseases[1,79–84]. In addition, we identify a unique Th subset that has features of both Treg and Th2 cells (Treg/Th2). Cells in this cluster express *Rora*, *Gata3* and *Il1r1* (encoding ST2), which were identified as markers of tissue adaptation for Tregs[48]. These cells also express high levels of *Ikzf2* which is linked to Treg stability[47,49]. These observations suggest that the majority of the Treg/Th2 cells are stable tissue adapted Tregs. Intriguingly, the pTh2 cell subsets show some similarities with the Treg/Th2 cell subset. Many marker genes previously attributed to lung pTh2 cells such as *Il1rl1* (ST2), *Areg* and *Bhlhe40* are not unique to the pTh2 cell subsets. Of note, previous studies in mice relied mostly on ST2 to define pTh2 cells[15,18,42,43,51,52], however, a detailed characterisation of lung ST2$^+$ Th subsets in response to HDM is lacking. Our scRNA-seq analysis suggests that using ST2 alone is insufficient to define pTh2 cells since it is also expressed by Treg/Th2 cells, which we collectively termed non-pTh2 ST2$^+$ Th cells due to their high expression of FoxP3. Notably, to exclude FoxP3-expressing ST2$^+$ Th cells and obtain "pure" ST2$^+$ pTh2 cells, FoxP3 reporter mice could be used as previously demonstrated[52]. However, due to the laborious nature of acquiring and maintaining reporter mice as well as possible background problems arising from crossing different types of mouse strains, using surface markers might represent a generally applicable and unbiased approach. These limitations underline the need to define markers to clearly distinguish the different ST2$^+$ Th subsets. Thus, we identify surface markers distinctively expressed by the ST2$^+$ Th subsets and establish a flow cytometry gating strategy for distinguishing each subset. We anticipate that by using our proposed surface marker panel, mouse lung pTh2 cell subsets can be distinguished from the non-pTh2 ST2$^+$ Th subsets and purified for detailed functional and phenotypic characterisation.

Our work further reveals the shared and distinct transcriptional signatures of peTh2 and Th2 Trm cells. Both subsets express high levels of transcriptional regulators such as *Gata3*, *Bhlhe40*, *Pparg*, *Gadd45b* and *Nfat5*. However, peTh2 cells exhibit more effector phenotype with increased expression of type 2 effector molecules than Th2 Trm cells. The reduced expression of these effector molecules in Th2 Trm cells suggests distinct differentiation stages between these two pTh2 cell subsets. For instance, Th2 Trm cells show a diminished expression of *Il13* mRNA despite expressing the protein, suggesting that they already down-regulated/modulated *Il13* on a transcriptional level and indicating that they are more differentiated than peTh2 cells. This is further substantiated by their acquisition of a tissue-residency program such as enhanced expression of *Cd69* and reduced expression levels of *S1pr1*, *Klf2* and *Ccr7*. Additionally, in contrast to peTh2 cells, lung Th2 Trm cells are enriched in *Zfp36l2*, *Lpar6*, *Slc38a2* and *Egr1*. *Zfp36l2* encodes zinc finger protein 36 like 2 (Zfp36l2), which is an RNA-binding protein (RBP). RBPs bind the adenine-uridine-rich elements (AREs) in the 3´ untranslated regions of many mRNAs to regulate gene expression[85] and are suggested to exhibit an anti-inflammatory role by restricting the expression of various proinflammatory molecules[86]. Interestingly, it has been demonstrated that *Zfp36l2* is essential to repress interferon-γ translation from pre-formed cytokine encoding RNA in mouse and human memory T cells, serving as a regulatory mechanism to limit aberrant cytokine secretion[87]. It is unclear what mechanism causes the diminished mRNA expression of effector molecules such as *Il13* in the Th2 Trm cell subset, but we speculate that *Zfp36l2* might play a role in this regulation to limit the excessive production of pathogenic molecules by these cells. Furthermore, the Th2 Trm cells are highly enriched for *Fos*, *Fosb* and *Jun*;

members of the AP-1 transcription factor (TF) family[88]. Increased expression and accessibility of these TFs in skin CD8$^+$ Trm cells have been reported recently. Notably, deletion of *Fos* and *Fosb* impaired skin CD8$^+$ Trm formation[89]. Thus, our analyses identify metabolic and transcriptional regulators that warrant further investigation, especially in Th2 Trm formation and maintenance.

Our understanding of the molecular mechanisms regulating pTh2 cells is hindered by the lack of an in vitro model to generate and study them in more detail. Our data uncover that co-stimulation of GITR together with TSLPR, induces pTh2 differentiation in vitro. We confirm that these in vitro generated pTh2 cells are highly similar to lung peTh2 cells and exhibit enhanced upregulation of pathogenic molecules and TFs. Therefore, our work unravels the importance of co-stimulatory factors like ligands of TNFRSF members together with TSLP to drive the differentiation of pTh2 cells. GITR ligand is indeed upregulated in the peripheral blood of asthmatic children and on lung dendritic cells of HDM-exposed mice, suggesting a critical role of the GITR-GITRL axis in allergic asthma[72]. Furthermore, using our in vitro model, we show that targeting p38 MAPK suppresses IL-5 and IL-13 expression in pTh2 cells. Our finding that the p38 MAPK pathway is crucial for pTh2 differentiation independently of IL-33 adds to previous studies demonstrating a role for this pathway in regulating IL-5 and IL-13 expression in both human and mouse T cells[15,90–92]. Since p38 MAPK has diverse effects and controls many physiological processes[93–95], inhibiting it may be deleterious. However, identifying its downstream targets and defining their role in pTh2 cell differentiation might potentially lead to better therapeutic targets for treating allergic asthma and other allergic diseases. We anticipate that our in vitro model for generating pTh2 cells will be leveraged to strengthen our understanding of the molecular processes driving pTh2-mediated allergic asthma.

It is well established that epigenetic regulators such as HDACs play important roles in T cell biology[30,31,38]. Our work highlights the importance of HDAC1 in HDM-induced allergic asthma and pTh2 cell differentiation. Deletion of HDAC1 in T cells promotes HDM-induced airway inflammation and the pathogenicity of lung pTh2 cell subsets. In particular, HDAC1 dampens IL-5 production. In addition, HDAC1 appears to restrain Th2 Trm cell formation, as evidenced by the increased proportion of Th2 Trm cells in absence of HDAC1. Several lines of evidence implicate Th2 Trm cells as important players in mediating allergic asthma due to their propensity to rapidly secrete effector molecules upon re-encounter with an allergen[15,41,51,52,56,57]. Thus, identifying the mechanisms underlying their generation and maintenance is of great importance for developing new therapies for allergic asthma[52]. Moreover, consistent with our in vivo findings, HDAC1 is also essential to limit the pathogenicity of in vitro generated pTh2 cells, particularly IL-5. These findings have broad implications, as natural HDAC inhibitors such as SCFAs, derived from bacterial fermentation of non-digestible dietary fibre[36,96–98], are regarded as immunosuppressive[99–101]. Subsequently, SCFAs are suggested as potential therapeutic agents to ameliorate various diseases including allergic asthma[102–108], at odds with the findings that they could augment type 2 cytokine production[109,110], as well as promote allergic asthma[110]. Therefore, it is crucial to further investigate the effects of natural HDAC inhibitors like SCFAs on pTh2 cell differentiation and whether they are suitable candidates for treating allergic asthma. Based on our findings, augmenting the activity of HDAC1 might represent an approach to limiting allergic asthma.

In addition, our data provide evidence that HDAC1 represses chromatin accessibility of the Th2 cytokine locus, and that pathogenic differentiation conditions lead to a partial attenuation of HDAC1 function either by disruption/re-localization of corepressor complexes[111], inhibition of HDAC1 activity via post-translational modifications[112], and/or a stronger recruitment/activation of histone acetyltransferases (HAT), overriding HDAC1 repression[111,113]. The inactivation of HDAC1 results in enhanced opening of the chromatin at the

Th2 cytokine locus. It is known that the transcription of Th2 cytokine genes is modulated by various regulatory elements such as the conserved GATA3 response element, DNase I hypersensitive sites, and the locus control region (LCR)[114–117]. These regions are docking sites for GATA3[114,118–120]. We observed increased chromatin accessibility at the type 2 cytokine gene loci particularly within the *Rad50* gene. This region contains an LCR that coordinates *Il5* gene transcription by enhanced chromatin interactions[117]. We speculate that the increase in chromatin accessibility of the Th2 cytokine locus in absence of HDAC1 along with the binding of a specific set of TFs such as PPARγ, RBPJ, NF-κB1, JUNB and BHLHE40 that are known regulators of Th2 cytokine expression[16,17,61,63,121–123], is likely the mechanism that drives the enhanced type 2 cytokine expression, particularly *Il5*, by pTh2 cells. Due to the complexity of gene regulation, we cannot exclude additional regulatory mechanisms such as post-transcriptional regulation and post-translational modification of transcription factors. Overall, we provide a mechanistic framework which highlights HDAC1 as a gatekeeper for Th2 cytokine expression, in particular Il5. During activation under pathogenic conditions, repression via HDAC1 is attenuated, preparing the chromatin for GATA3 binding. As GATA3 has been shown to facilitate chromatin loop formation[117], we hypothesise that enhanced binding of GATA3 to regulatory elements in the Th2 cytokine locus will result in a more stable 3D chromatin structure and an increase in Th2 cytokine production. Experiments using this pTh2 protocol to investigate this mechanism at 3D level in even more molecular detail are warranted.

Collectively, we resolve the heterogeneity of mouse lung pTh2 cell subsets and reveal their transcriptional and phenotypic profiles. We identify a system to generate pTh2 cells in vitro, which we used to delineate the molecular mechanisms critical for their pathogenicity. Our work further illustrates that HDAC1 is a master regulator of pTh2 cells, suggesting that therapeutic approaches aimed at inhibiting HDACs in allergic asthma, like the use of SCFAs and synthetic HDAC inhibitors, should be revisited. An in-depth understanding of how pTh2 cells are regulated will be crucial for developing novel therapies for pTh2-mediated allergic diseases including allergic asthma, allergic rhinitis, atopic dermatitis, eosinophilic esophagitis and food allergy.

## Methods

### Mice
We used Hdac1[flox/flox] (HDAC1[f/f])[38], CD4-Cre[124] and IL-13tdTomato[39] mice on C57BL/6 J background. Mice were kept at the Core Facility Laboratory Animal Breeding and Husbandry (CFL) at the Medical University of Vienna under controlled, standardised conditions (artificial reverse day-night cycle 12:12; room temperature $22 \pm 2\,°C$, humidity $55 \pm 10\%$). The animals received autoclaved complete food for mice (mouse husbandry, autoclavable; LASQCdiet® Rod16, Auto, 10 mm, Zero and Cert*), and water ad libitum. The hygiene status of the mice was regularly checked according to FELASA criteria. The animals were kept in type II IVC racks. Animal husbandry and experiments were reviewed and approved by the Institutional Review Board of the Medical University of Vienna and approved by the Austrian Ministry of Economy and Science (BMWFW-2020-0.547.902) and performed as per the guidelines of the Federation of European Laboratory Animal Science Associations. Mice were euthanised for experiments by neck dislocation.

### HDM model of allergic asthma
Eight to twelve weeks old female WT and HDAC1-cKO mice were sensitised intratracheally with 10 µg HDM (Greer Laboratories; Item no. XPB91D3A25, Lot no. 325470) in 40 µl phosphate buffer saline (PBS) at days 0 and 5. On days 12 and 14, the mice were challenged intranasally with 25 µg HDM in 40 µl PBS and euthanised on day 16. Control mice received 40 µl PBS alone during the sensitisation and challenge periods. All intratracheal and intranasal injections were performed under

anaesthesia by intraperitoneal injection with ketamine plus xylazine or light inhalation of isoflurane, respectively.

### BAL collection
To analyse immune cell infiltration in the lungs, BAL was obtained by flushing the lungs of each mouse three successive times with 1 ml PBS using a tracheal cannula. BAL samples were centrifuged, and red blood cell lysis was performed using 1x RBC lysis buffer (BioLegend, 420301). Total cells were determined using a Coulter counter (Beckman Coulter). The remaining cells were stained to identify eosinophils and other BAL cells by flow cytometry[125].

### Isolation of lung cells
To prepare single-cell suspension from the lungs, each lung was perfused with PBS via the right ventricle, and then harvested. The lungs were minced and digested in RPMI-1640 (Sigma-Aldrich, R8758) containing 5% FCS (Biowest, S181H), 150 U/ml collagenase type I (Gibco, 17100-017) and 50 U/ml DNase I (Sigma-Aldrich, DN25) for 1 h at 37 °C (with intermittent shaking every 15 min). A single-cell suspension was obtained by meshing digested lung tissue through a 70 µm cell strainer followed by washing with 2% FCS in PBS (FACS buffer). Red blood cell lysis was performed followed by washing and filtering the samples using a 40 µm strainer. The resulting single-cell suspension was used for further analysis.

### Flow cytometry
Antibodies were purchased from eBioscience: TCRβ (H57-597), ST2 (RMST2-2), FoxP3(FJK-16s) and IL-13(eBio 13 A); BioLegend: Ly6G (1A8), CD19 (6D5), CD11b (M1/70), MHC class II (AF6-120.1), Siglec F (S17007L), F4/80 (BM8), CD11c (N418), CD4 (RM4-5), CD44 (IM7), KLRG1 (2F1/KLRG1), CD27 (LG.3A10), GATA3 (16E10A23), CD69 (H1.2F3), PD1 (29 F.1A12), IL-4 (11B11), IL-5 (TRFK5), IL-9 (RM9A4), RANKL (IK22/5) and GM-CSF (MP1-22E9); and BD Biosciences: CD62L (MEL-14), CD25 (PC61), CD8α (53-6.7), GATA3 (L50-823) and CD44 (IM7). For antibody details (clone, dilution, vendor) see Supplementary Data 12.

For the extracellular staining of lung and BAL cells, the cells were preincubated with anti-CD16/32 (Clone: 2.4G2, BD Biosciences, 553142) for 2 min to block FC receptors and prevent non-specific binding, followed by surface staining with a fixable viability dye (eBioscience) and antibodies. Extracellular staining was performed at 4 °C for 30 min, then followed by a washing step with FACS buffer, centrifugation and analysis, or intracellular staining for cytokine or transcription factor analysis.

For analyses requiring intracellular cytokine staining, cells were fixed with 100 µl fixation buffer (BioLegend, 420801) for 15 min, then washed with FACS buffer and permeabilised with 1X permeabilization buffer (BioLegend, 421002) for 45 min, without washing, antibodies were added directly into the tubes. Cells were incubated for another 45 min, then washed and analysed. The FoxP3 staining buffer kit (eBioscience, 00-5523-00) was used for ex vivo staining of transcription factors and used following the manufacturer's instructions. Stained cells were measured using LSRFortessa (BD Biosciences) and analysed by FlowJo (v.10.8.1, Tree Star)

### Ex vivo restimulation of lung cells
To detect cytokines in lung T cells, $6 \times 10^6$ lung cells were plated in a 6-well plate (Sarstedt, 83.3920) and restimulated with 50 ng/ml Phorbol 12-myristate 13-acetate (PMA) and 750 ng/ml ionomycin (both from Sigma-Aldrich) in the presence of GolgiPlug and GolgiStop (both from BD Biosciences) for 4 h at 37 °C. All cells were cultured in complete RPMI-1640 (RPMI-1640 supplemented with 10% FCS (Biowest, S181H), penicillin (100 U/ml) plus streptomycin (100 µg/ml) (Sigma-Aldrich, P0781), 2 mM Glutamax (Thermo Fisher Scientific, 35050038), 55 µM β-Mercaptoethanol (Sigma-Aldrich, M6250)). Intracellular cytokine staining was performed as described above and samples were measured by flow cytometry.

## Single-cell analysis of lung CD4⁺ T cells

**Sample preparation.** Single-cell suspensions from lungs of WT and HDAC1-cKO IL-13 tdTomato reporter mice sensitised and challenged with HDM or PBS were extracellularly stained with a fixable viability dye and the following anti-mouse antibodies: CD19, CD8α, TCRβ, CD4, CD62L, CD44 and CD25. Cells were incubated at 4°C for 30 min, then washed and centrifuged. Followed by cell sorting using BD FACSAria Fusion to sort for different CD4⁺ T cell populations. To obtain the populations of interest, we first excluded all CD19 (B cells) and CD8α (CD8⁺ cytotoxic T cells) positive cells and retained the TCRβ⁺CD4⁺ T cells. Within the CD4⁺ T cells, we further excluded the CD25$^{high}$ cells to enrich for non-Treg effector cells. We then sorted the following CD4⁺ T cell populations: (1) naïve (CD62L⁺CD44⁻IL-13tdTomato⁻), (2) IL-13⁻ Th cells (CD62L⁻CD44⁺IL-13tdTomato⁻) and (3) IL-13⁺ Th cells (CD62L⁻CD44⁺IL-13tdTomato⁺). Naïve and IL-13⁻ Th cells were sorted from WT and HDAC1-cKO mice exposed to either PBS or HDM, while IL-13⁺ Th cells were sorted only from WT and HDAC1-cKO mice exposed to HDM. We obtained a total of ten samples based on the two genotypes (WT and HDAC1-cKO) and experimental conditions (HDM and PBS) (see Fig. 1a). We pooled lung cells from three mice per group to enable us to sort sufficient number of cells. After sorting the populations of interest, we centrifuged the samples and resuspended them in 100 μl FACS buffer then incubated each sample with a unique hashtag oligonucleotide (HTO) antibody[126], using 0.5 μl HTO antibody (BioLegend) per sample. The samples were incubated at 4°C for 30 min, followed by two washing steps, first with FACS buffer followed by 1 ml PBS containing 0.04% bovine serum albumin (BSA, Sigma-Aldrich). The samples were then resuspended in 100 μl PBS containing 0.04% BSA, and cells were counted. 3000 cells were taken from each sample and pooled. A total of 30,000 cells (from 10 samples) were submitted for each scRNA-seq experiment. Two independent experiments with 10 samples each were performed and data integrated for the analysis. Each sample is a pool of cells from three different experimental animals per group.

**Library preparation and sequencing.** Single-cell suspensions were counted and diluted according to manufacturer recommendations before loading onto a Chromium Controller (10x Genomics, Pleasanton, CA, USA). Single-cell RNA-seq libraries were generated using the Next GEM Single Cell 3′ Reagent Kit (v3.1, 10x Genomics, Pleasanton, CA, USA) according to the manufacturer's instructions aiming for a maximum cell recovery of 10,000 cells. NGS library concentrations were quantified with the Qubit 2.0 Fluorometric Quantitation system (Life Technologies, Carlsbad, CA, USA) and the size distribution was assessed using the 2100 Bioanalyzer instrument (Agilent, Santa Clara, CA, USA) before sequencing on a NovaSeq 6000 instrument (Illumina, San Diego, CA, USA). Both sequencing and data processing using the Cell Ranger suite (6.1.2, 10X Genomics, Pleasanton, CA, USA) was performed by the Biomedical Sequencing Facility at the CeMM Research Centre for Molecular Medicine of the Austrian Academy of Sciences.

**Analysis of scRNA-seq data.** Subsequent analyses including demultiplexing, quality control, processing, integration and visualisation were performed in R (v.4.1.2) using the Seurat package (v.4.0.6)[127]. For each dataset, HTO counts were first normalised using centred log-ratio (CLR) transformation, followed by demultiplexing of the cells based on their origin (HTO labelling) using HTODemux. After demultiplexing, we excluded the negative cells and doublets and retained only the single cells for downstream analyses. Next, quality control was performed to exclude low-quality cells by excluding cells with more than 10% mitochondrial genes, and less than 200 or greater than 7500 features (genes). We then performed normalisation, feature selection and data scaling. Each dataset was normalised using the NormalizeData function based on the default parameters (using the

"logNormalize" method and a scale factor of 10,000). Highly variable genes were identified using the FindVariableFeatures function (using the "vst" method and top 2000 variable genes) and transformed the data with the ScaleData function. Next, principal components analysis (PCA) was performed with RunPCA to determine the dimensionality of the data followed by clustering of the cells using the FindNeighbors (on the first 20 principal components) and FindClusters (at a resolution of 1) functions. We then performed an additional step of doublet exclusion using DoubletFinder (v.2.0.3)[128], assuming a 7.6% doublet formation rate as recommended by 10x Genomics. After removing the doublets, we retained the singlets and performed normalisation, feature selection, data scaling, dimensionality reduction and clustering, as previously mentioned. After processing each dataset separately, we merged them and performed normalisation, feature selection, data scaling, dimensionality reduction and clustering, as previously mentioned. We then integrated both datasets by canonical correlation analysis (CCA) using the FindIntegrationAnchors and IntegrateData functions[129], then scaled the integrated data, performed dimensionality reduction, followed by clustering of the cells using FindNeighbors (on the first 15 principal components) and FindClusters (at a resolution of 0.5). The FindAllMarkers and FindMarkers functions were used to identify DEGs by only including genes that are expressed in at least 25% of the cells and with a minimum log fold change of 0.25 (min.pct = 0.25, logfc.threshold = 0.25). Statistical significance was determined using the default Wilcoxon rank sum test with Bonferroni correction. In addition to Seurat's visualisation tools, ggplot2 (v.3.4.0)[130], EnhancedVolcano (v.1.12.0)[131], scCustomized[132], and DeepVenn [http://www.deepvenn.com/] were also used. Gene set enrichment analysis[133] was performed in R using the fgsea package (v.1.20.0)[134]. The genes were pre-ranked based on avg_log2FC. To determine enriched pathways, we used the mouse HALLMARK gene set collection in The Molecular Signatures Database (mSigDB) (v.7.5.1)[135]. The stat_compare_means (using the Wilconxon test) function of the ggpubr package (v 0.6.0) was used to derive P-values for the comparison of selected genes of interest between WT and HDAC1-cKO cells in a specific cluster.

## Mouse T helper cell differentiation

Single-cell suspensions were prepared from spleen and lymph nodes from male or female 6- to 12-week-old mice, then naïve CD4⁺ T cells (CD4⁺CD62L⁺CD44⁻) were isolated using the Miltenyi Naïve CD4⁺ T cell isolation kit (Miltenyi Biotec, 130-104-453), with some modifications: cells were subjected to red blood cell lysis prior to isolation with the kit, and washing steps were added between each incubation times. The purity was monitored by flow cytometry. Isolated naïve CD4⁺ T cells (1 ×10⁵/ml) were plated in a 48-well plate (Sarstedt, 83.3923) precoated overnight at 4 °C with 1 μg/ml anti-CD3ε (clone: 145-2C11, 553057) and 3 μg/ml anti-CD28 (clone: 37.51, 553294) both from BD Biosciences, and cultured in complete RPMI with different polarisation conditions. The following conditions were used for Th2 cell differentiation: IL-4 (25 ng/ml, Peprotech, 214-14), IL-2 (10 U/ml, Peprotech, AF-200-02), anti-IFN-γ (10 μg/ml, clone XMG1.2, BioXCell, BE0055) and anti-TGF-β1 (5 μg/ml, clone 1D11.16.8, BioXCell, BE0057). Th17: IL-6 (20 ng/ml, BioLegend, 575702), TGF-β1 (1 ng/ml, BioLegend, 580702). To investigate the induction of a pathogenic program in Th2 cells, naïve CD4⁺ T cells were cultured under Th2-promoting conditions together with TSLP (50 ng/ml, R&D, 555-TS-010) and agonistic antibodies for GITR (5 μg/ml, clone DTA-1, BioXCell, BE0063), OX40 (5 μg/ml, clone OX-86, BioXCell, BE0031) or 4-1BB (5 μg/ml, clone 3H3, BE0239) all from BioXCell. Unless otherwise stated, CD4⁺ T cells were cultured for 5 days, and on day 3 of culture, cells were passaged (1:2) in complete RPMI-1640 containing only anti-IFN-γ (10 μg/ml) and anti-TGF-β (5 μg/ml) without any activating or polarising conditions. To assess the role of AP-1 and mitogen-activated protein kinases (MAPKs) pathways, naïve CD4⁺ T cells were cultured for 5 days under Th2-promoting conditions as above, and under pathogenic Th2 (pTh2)-promoting

conditions: IL-4 (25 ng/ml), IL-2 (10 U/ml), TSLP (50 ng/ml), DTA-1 (5 μg/ml), anti-IFN-γ (10 μg/ml) and anti-TGF-β1 (5 μg/ml) and pTh2-promoting condition together with inhibitors for AP-1 (10 μM, T-5224, HY-12270), p38 (10 μM, SB 203580, HY-10256), Erk1/2 (1 μM, U0126-EtOH, HY-12031) or JNK (1 μM, SP600125, HY-12041) all from Med-ChemExpress. On day 3, cells were split (1:2) as indicated above. The inhibitors were added at the beginning of the experiments. For cytokine measurement, cultured cells were restimulated on day 5 with 50 ng/ml Phorbol 12-myristate 13-acetate (PMA) and 750 ng/ml ionomycin (both from Sigma-Aldrich) in the presence of GolgiPlug and GolgiStop (both from BD) for 4 h at 37 °C. Cells were then harvested, stained, and analysed by flow cytometry.

### RNA-seq analysis of in vitro generated cells

**Sample and library preparation for RNA-seq.** Naïve CD4+ T cells were isolated as previously described. Cells ($1 \times 10^5$/ml) were plated in a 48-well plate (Sarstedt, 83.3923) precoated overnight at 4 °C with 1 μg/ml anti-CD3ε and 3 μg/ml anti-CD28. Then cultured in complete RPMI under Th2-promoting conditions: IL-4 (25 ng/ml), IL-2 (10 U/ml), anti-IFN-γ (10 μg/ml) and anti-TGF-β1 (5 μg/ml) or pTh2-promoting conditions: IL-4 (25 ng/ml), IL-2 (10 U/ml), TSLP (50 ng/ml), DTA-1 (5 μg/ml), anti-IFN-γ (10 μg/ml), and anti-TGF-β1 (5 μg/ml) for 3 days. On day 3, cells were harvested, and total RNA was isolated using the RNeay Mini kit (Qiagen) according to the manufacturer's instructions.

The amount of total RNA was quantified using the Qubit 2.0 Fluorometric Quantitation system (Thermo Fisher Scientific, Waltham, MA, USA) and the RNA integrity number (RIN) was determined using the 2100 Bioanalyzer instrument (Agilent, Santa Clara, CA, USA). RNA-seq libraries were prepared with the TruSeq Stranded mRNA LT sample preparation kit (Illumina, San Diego, CA, USA) using Sciclone and Zephyr liquid handling workstations (PerkinElmer, Waltham, MA, USA) for pre- and post-PCR steps, respectively. Library concentrations were quantified with the Qubit 2.0 Fluorometric Quantitation system (Life Technologies, Carlsbad, CA, USA) and the size distribution was assessed using the 2100 Bioanalyzer instrument (Agilent, Santa Clara, CA, USA). For sequencing, samples were diluted and pooled into NGS libraries in equimolar amounts.

**Sequencing and raw data acquisition.** Expression profiling libraries were sequenced on a HiSeq 4000 instrument (Illumina, San Diego, CA, USA) following a 50-base-pair, single-end recipe. Raw data acquisition (HiSeq Control Software, HCS, HD 3.4.0.38) and base calling (Real-Time Analysis Software, RTA, 2.7.7) were performed on-instrument, while the subsequent raw data processing off the instruments involved two custom programs based on Picard tools (2.19.2). In the first step, base calls were converted into lane-specific, multiplexed, unaligned BAM files suitable for long-term archival (IlluminaBasecall-sToMultiplexSam, 2.19.2-CeMM). In a second step, archive BAM files were demultiplexed into sample-specific, unaligned BAM files (IlluminaSamDemux, 2.19.2-CeMM).

**Analysis of RNA-seq data.** NGS reads were mapped to the Genome Reference Consortium GRCm38 assembly via "Spliced Transcripts Alignment to a Reference" (STAR, 2.7.5a)[136] utilising the "basic" Ensembl transcript annotation from version e100 (April 2020) as reference transcriptome. Since the mm10 assembly flavour of the UCSC Genome Browser was preferred for downstream data processing with Bioconductor packages for entirely technical reasons, Ensembl transcript annotation had to be adjusted to UCSC Genome Browser sequence region names. STAR was run with options recommended by the ENCODE project[137]. NGS read alignments overlapping Ensembl transcript features were counted with the Bioconductor (3.11) GenomicAlignments (1.24.0) package via the summarizeOverlaps function in Union mode, ignoring secondary alignments and alignments not passing vendor quality filtering. Since the Illumina TruSeq stranded

mRNA protocol leads to the sequencing of the first strand, all alignments needed inverting before strand-specific counting in feature (i.e., gene, transcript and exon) orientation. Transcript-level counts were aggregated to gene-level counts and the Bioconductor DESeq2 (1.28.1) package[138] was used to test for differential expression based on a model using the negative binomial distribution. Biologically meaningful results were extracted from the model, log2-fold values were shrunk with the ashr (2.2.-47) package[139], while two-tailed *P*-values obtained from Wald testing were adjusted with the bioconductor independent hypothesis weighting (IHW, 1.16.0) package[140]. The resulting gene lists were annotated and filtered for significantly differentially up-and down-regulated genes based on adjusted *P*-value < 0.1. Plots were generated using ggplot2 (v.3.4.0) and EnhancedVolcano (v.1.12.0) and pheatmap (v.1.0.12). GSEA was performed in R using the fgsea package (v.1.20.0). The genes were pre-ranked based on log2FC. Pathway analysis was performed using the MSigDB (v.7.5.1) mouse HALLMARK gene set collection.

### Proteomics analysis of in vitro generated cells

**Cell and mass spectrometry sample preparation.** Th2 and pTh2 cells were cultured as in the RNA-seq analysis. On day 3, the cells were harvested, washed and pelleted before lysis.

Cells were resuspended in lysis buffer (8 M urea (VWR, 0568-500 G), 50 mM Tris-HCl pH8.0, 150 mM NaCl, 1 mM PMSF, 5 mM sodium butyrate (Sigma-Aldrich,303410), 10 ng/ml NAM (Sigma Aldrich, 72340), 10 ng/ml trichostatin A (TSA) (Sigma Aldrich, T8552-1MG), 1x cOmplete protease inhibitor cocktail (+ EDTA) (Roche, 11697498001) and 250 U/replicate benzonase (Merck, 1.01695.0001)), lysed using a Bioruptor sonication device (Diagenode) (settings: 30 s sonication (power level H, 30 s cooling, 5 cycles), centrifuged for 10 min at $15{,}000 \times g$, 4 °C and precipitated with 4× volumes of cold (−20 °C) 100% acetone. Protein samples were washed with 80% acetone (−20 °C), air dried, dissolved in 8 M urea, 50 mM ABC (Sigma Aldrich, 09830-500 G), reduced in 10 mM DTT (Roche, 10197777001) for 45 min (RT), carbamidomethylated with 20 mM IAA (Sigma Aldrich, I6125-5G) for 30 min at RT in the dark, and quenched with 5 mM DTT for 10 min. The urea concentration was reduced to 4 M using 50 mM ABC. Samples were pre-digested with Lys-C (FUJIFILM Wako Pure Chemical Corporation, 125-02543) for 90 min at 37 °C (enzyme-to-substrate ratio of 1:100) and subsequently digested overnight at 37 °C with trypsin (Trypsin Gold, Mass Spec Grade, Promega, V5280) at an enzyme-to-substrate ratio of 1:50. Digests were stopped by acidification with TFA (Thermo Scientific, 28903) (0.5% final concentration) and desalted on a 50 mg tC18 Sep-Pak cartridge (Waters, WAT054960). Peptide concentrations were determined and adjusted according to UV chromatogram peaks obtained with an UltiMate 3000 Dual LC nano-HPLC System (Dionex, Thermo Fisher Scientific), equipped with a monolithic C18 column (Thermo). Desalted samples were lyophilised overnight.

**Isobaric labelling using TMTpro 16plex.** 300 μg of trypsin-digested and desalted peptides were used for each sample for isobaric labelling. Lyophilized peptides were dissolved in 20 μl 100 mM triethylammonium bicarbonate (TEAB) (Supelco, 18597-100 ML). 500 μg of each TMT 10plex reagent (Thermo Scientific, 90110) were dissolved in 30 μl of acetonitrile (ACN) (VWR, 83639.320) and added to the peptide/TEAB mixes. Samples were labelled for 60 min at RT and small aliquots were checked for labelling efficiency. Reactions were quenched using hydroxylamine (Sigma Aldrich, 438227-50 ML) (0.4% final concentration), pooled, desalted using Sep-Pak tC-18 (200 mg) cartridges (Waters, WAT054925), dried in a SpeedVac vacuum centrifuge (Eppendorf), and subsequently lyophilized overnight. Neutral pH fractionation was performed using a 60 min gradient of 4.5 to 45% ACN (VWR, 83639.320) in 10 mM ammonium formate (1 ml formic acid (26 N) (Merck, 1.11670.1000), 3 ml ammonia (13 N) (1.05432.1000) in

300 ml $H_2O$, pH = 7–8, dilute 1:10) on an UltiMate 3000 Dual LC nano-HPLC System (Dionex, Thermo Fisher Scientific) equipped with an xBridge Peptide BEH C18 (130 Å, 3.5 µm, 4.6 mm × 250 mm) column (Waters) (flow rate of 1.0 ml/min). Fractions were collected and subsequently pooled in a non-contiguous manner into 8 pools and lyophilized overnight.

**Mass spectrometry measurements.** DDA-SPS-MS3 method with online real-time database search (RTS) was performed on an UltiMate 3000 Dual LC nano-HPLC System (Dionex, Thermo Fisher Scientific), containing both a trapping column for peptide concentration (PepMap C18, 5×0.3 mm and 5 µm particle size) and an analytical column (PepMap C18, 500 x 0.075 µm, 2 µm particle size, Thermo Fisher Scientific), coupled to an Orbitrap Eclipse mass spectrometer (Thermo Fisher) via a FAIMS Pro Duo ion source (Thermo Fisher). The instrument was operated in data-dependent acquisition (DDA) mode with dynamic exclusion enabled. For peptide separation on the HPLC, the concentration of organic solvent (acetonitrile, VWR, 83639.320) was increased from 1.6% to 32% in 0.1% formic acid at a flow rate of 230 ml/min, using a 2 h gradient time for proteome analysis. Peptides were ionised with a spray voltage of 2,4 kV. The instrument method included Orbitrap MS1 scans (resolution of 120,000; mass range 375–1500 m/z; automatic gain control (AGC) target 4e5, max injection time of 50 ms, FAIMS CV -40V, -55V and -70V, dynamic exclusion 45 sec), and MS2 scans (CID collision energy of 30%; AGC target 1e4; rapid scan mode; max injection time of 50 ms, isolation window 1.2 m/z). RTS was enabled with full trypsin specificity, max. 2 missed cleavages, max. search time of 40 ms, max. 2 variable modifications, max. 10 peptides per protein, and considering the following modifications: Carbamidomethyl on cysteine and TMTpro16plex on Lys and peptide N-terminus as static, oxidation on Met and deamidation on Asn and Gln as variable modifications. The scoring thresholds were applied (1.4 for Xcorr, 0.1 for dCn, 10 ppm precursor tolerance) and the *Mus musculus* (mouse) Uniprot database (release 2020.01) was used for the search. Quantitative SPS-MS3 scans were performed in the Orbitrap with the following settings: HCD normalised collision energy 50%, resolution 50,000; AGC target 1.5e5; isolation window 1.2 m/Z, MS2 isolation window 3 m/z, 10 notches, max injection time of 150 ms. The total cycle time for each CV was set to 1.2 s.

**Mass spectrometry data analysis.** MS raw files were split according to CVs (−40V, −55V, −70V) using FreeStyle 1.7 software (Thermo Scientific). Raw MS data were analysed using MaxQuant[141] software version 1.6.17.0, using default parameters with the following modifications. MS2 spectra were searched against *Mus musculus* (mouse) Uniprot database (release 2021.03; with isoforms) and a database of common laboratory contaminants. Enzyme specificity was set to "Trypsin/P", the minimal peptide length was set to 7 and the maximum number of missed cleavages was set to 2. Carbamidomethylation of cysteine was searched as a fixed modification. "Acetyl (Protein N-term)", "Oxidation (M)" were set as variable modifications. A maximum of 5 variable modifications per peptide was allowed. The identification and quantification information of proteins was obtained from the MaxQuant "ProteinGroups" tables. Data were analysed in R (4.1.0) using custom scripts. The analysis procedure covered: correction for isotopic impurities of labels, within-plex median normalisation, and statistical analysis between-group comparisons using LIMMA (3.50)[142]. Plots were generated using ggplot2 (v.3.4.0) and EnhancedVolcano (v.1.12.0), and pheatmap (v.1.0.12)[143].

**ATAC-seq analysis of in vitro generated cells**
**High-throughput chromatin accessibility mapping library preparation.** Th2 and pTh2 cells were cultured as in the RNA-seq analysis for 24 and 48 h. High-throughput chromatin accessibility mapping (ATAC-seq) was performed as previously described, with minor modifications[76,144]. For each experiment, a maximum of 50,000 cells were collected at 300 x g for 5 min at 4°C. After centrifugation, the cell pellet was carefully resuspended in transposase reaction mix (12.5 µl 2 x TD buffer, 2 µl TDE1 (Illumina, San Diego, CA, USA), 10.25 µl nuclease-free water, and 0.25 µl 1 % digitonin (Promega, Madison, WI, USA)) for 30 min at 37 °C. Following DNA purification (MinElute PCR Purification kit, QIAGEN N.V, Venlo, Netherlands) and eluting in 11 µl, 1 µl eluate was used in a quantitative PCR (qPCR) reaction to estimate the optimal number of cycles for library amplification. The remaining 10 µl of each tagmented sample was then amplified corresponding to the $C_q$ value (i.e., the cycle number at which fluorescence has increased above background levels) in the presence of custom Nextera primers. PCR amplification was followed by SPRI (Beckman Coulter, Brea, CA, USA) size selection to exclude fragments larger than 1200 bp. The DNA concertation of each library was assessed with a Qubit 2.0 Fluorometric Quantitation system (Life Technologies, Carlsbad, CA, USA), before pooling in equimolar amounts. The resulting pools were sequenced on a NovaSeq 6000 instrument (Illumina, San Diego, CA, USA) in a 100-base pair paired-end configuration. Chromatin accessibility mapping by ATAC-seq was done in three biological replicates.

**High-throughput chromatin accessibility mapping analysis.** NGS reads in unaligned BAM files were converted into FASTQ format with samtools[145], NGS adaptor sequences were removed via fastp[146] (0.23.2, GTCTCGTGGGCTCGG) and the reads were aligned to the GRCm38 (UCSC Genome Browser mm10) assembly with Bowtie2[147] (2.4.4, –very-sensitive, –no-discordant, –maxins 2000), before deduplicating with samblaster[148] (0.1.24). BED-files were generated from the BAM-files using bedtools bamtobed, and further into a file compatible with scATAC-seq analysis software (atac_fragments.tsv.gz) using a custom Java program (https://github.com/henriksson-lab/bulkatac2fragments). For speed of calculation, only chromosome 11 was used to represent global chromatin activity. Signac 1.13 was used for dimensional reduction, visualisation and differential accessibility testing[149]. Peak calling was performed via built-in MACS2[150]. Motifs were called using the JASPAR2020 database, and motif activity was computed using chromVAR (https://www.nature.com/articles/nmeth.4401). The motif activity correlation (difference), pTh2 vs Th2, were calculated using Rfast::correls.

To compute differentially expressed genes to compare with motif activity correlation, we first subsetted the scRNA-seq count data based on hashtag oligos to exclude naive populations (HDAC1cKO-HDM-naive, HDAC1cKO-PBS-naive, WT-PBS-naive and WT-HDM-naive). We then used Seurat FindMarkers on the peTh2 cluster vs all other clusters combined, to find differentially expressed genes.

**Statistical analysis**
Statistical analysis of non-omics data was performed in GraphPad Prism (v.9.3.1). Statistical significance between two groups was determined by a two-tailed unpaired Mann-Whitney $U$ test. One-way or two-way ANOVA with Tukey's multiple comparisons test was used to compare more than two groups.

**Reporting summary**
Further information on research design is available in the Nature Portfolio Reporting Summary linked to this article.

## Data availability
Raw single-cell sequencing data generated in this study have been deposited in the ArrayExpress database under accession code #E-MTAB-13089 [https://www.ebi.ac.uk/biostudies/arrayexpress/studies/E-MTAB-13089?query=%23E-MTAB-13089]. RNA-seq data and ATAC-seq data generated in this study have been deposited in Gene Expression Omnibus (GEO) under accession code GSE235803 and

GSE280390 respectively. The mass spectrometry proteomics data have been deposited to the ProteomeXchange Consortium via the PRIDE[151] partner repository with the dataset identifier PXD042975. All other data supporting the findings of this research are available in the article and the Supplementary Material or from the corresponding author upon request. Source data are provided with this paper.

## Code availability
Custom R-scripts used in the proteomics analysis have been deposited to the ProteomeXchange Consortium via the PRIDE[151] partner repository with the dataset identifier PXD042975. R code for scRNA-seq and ATAC-seq analysis is available at GitHub (https://github.com/henriksson-lab/matarr2024). Software for converting bulk ATAC-seq to single-cell compatible format is available at (https://github.com/henriksson-lab/bulkatac2fragments).

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

## Acknowledgements

We thank the Biomedical Sequencing Facility at CeMM for assistance with next-generation sequencing and Michael Schuster for initial data processing and analysis of sequencing data. Mice were kept at the Core facility laboratory animal breeding and husbandry of the Medical University of Vienna. We also thank Dieter Printz of the FACS Core Unit at St. Anna CCRI for cell sorting. We are thankful to Shinya Sakaguchi for critical reading of the manuscript. We are grateful to Sahar Kazemi and Michelle Epstein for useful discussions on animal experiments and conceptual guidance in the allergy research field. This study has been funded by Austrian Science Foundation (FWF) through projects P30885 and F7004 (N.B.). J.H. is supported by Vetenskapsrådet grant numbers #2021-06602, #2024-03952 and Swedish Cancer Society#23 3102 Pj. This research was funded in part by the Austrian Science Fund (FWF) [DOI:10.55776/F70]. For open access purposes, the author has applied a CC BY public copyright license to any author accepted manuscript version arising from this submission.

## Author contributions

M.K. and N.B. conceptualised the study. M.K. designed, performed experiments, and analysed data including bioinformatics. M.A. contributed to cell culture and all animal experiments. T.K. and L.D. contributed to the planning of scRNA-seq experiments. L.D., M.M., M.W. assisted with bioinformatics analysis. W.R. and M.H. performed quantitative mass spectrometry and data analysis. W.R., T.K., L.D., C.B., M.H., W.E. and J.H. were involved in the experiment discussions and design. J.H. performed and supervised the scRNA-seq and performed ATAC-seq analysis. M.K. and N.B. wrote the manuscript with contributions from all co-authors.

## Competing interests

C.B. is a cofounder and scientific advisor of Myllia Biotechnology and Neurolentech. The remaining authors declare no competing interests.
