## [Transparent Peer Review file · Nature Communications]

Single-cell profiling uncovers regulatory programs of pathogenic Th2 cells in allergic asthma

Corresponding Author: Dr Nicole Boucheron

Version 0:

Reviewer comments:

Reviewer #1

(Remarks to the Author)
Summary of findings

In this novel and interesting study, the authors employed a mouse model of HDM exposure to characterize Th2-memory responses in the lung and dissect the role of HDAC1 in the regulation of HDM-induced airway inflammation. They found that HDAC1-cKO mice had exaggerated eosinophilic airways inflammation and enhanced IL-5 and IL-13 responses in ex vivo stimulated T-helper cells. They then employed IL-13 tdTomato reporter mice to isolate IL-13 positive T-helper cells, non-IL-13 producing T-helper cells, and naïve T-helper cells for single cell profiling. They showed that the T cell population was divided into 9 distinct clusters. The cells in cluster 2 expressed the highest levels of Th2 effector molecules (IL-4, IL-5, IL-13, Areg) and accordingly were designated pathogenic effector Th2 (peTh2). The cells in cluster 5 expressed moderate levels of Th2 effectors in combination with the TRM marker CD69, and accordingly were designated Th2 Trm cells. GSEA analysis demonstrated that the T-helper subset had shared overlapping gene signatures, and therefore the authors established a flow cytometry to better define the subsets. They showed that peTh2 cells can be identified as ST2+CD27-KLRG1-CD69lowPD1high, and Th2 Trm cells as ST2+CD27-KLRG1-CD69highPD1high.

The authors then set out to establish an in vitro model to study the differentiation of pTh2 cells. Given that the pTh2 cells express GITR, the authors differentiated naïve CD4 T cells in Th2 promoting conditions, with or without TSLP and/or a GITR agonist. They found that the combination of TSLP and the GITR agonist resulted in the highest production of Th23 effector cytokines, and this was further augmented in the HDAC1-cKO cells. The authors also performed bulk RNA-Seq to demonstrate that the in vitro pTh2 cells had a similar molecular signature to lung derived pTh2 cells.

Finally, the authors found that the lung derived and in vitro pTh2 cells were enriched with components of the AP-1, MAPK, and NFkB signaling pathways, and they employed the small molecule inhibitors to target AP-1 (T-5224) and p38 MAPK (SB 203580) on the in vitro differentiation of pTh2 cells. The found that SB 203580 markedly suppressed IL-5 and IL-13 responses but not IL-4 or gata-3 levels in WT and HDAC1-cKO pTh2 cells. In contrast, AP-1 inhibition with T-5224 had not effect on pTh2 differentiation. The authors conclude that their findings resolve the heterogeneity of murine lung pTh2 cell subsets and unveil their transcriptional and phenotypic profiles, identify HDAC1 is a master regulator of pTh2 cells, and establish a novel in vitro model to generate pTh2 cells in vitro.

Major comments

1. The findings in this manuscript are largely based on a single cell profiling experiment consisting of 10 samples. The experiment does not contain any biological replicates. To ensure the reproducibility and robustness of the findings, it is important to replicate the experiment.
2. The single cell clusters were annotated using a series of known marker genes selected by the authors. It is important to confirm the cell annotations using an unbiased reference-based approach such as Azimuth.
3. The Th2-tissue resident memory cells were defined primarily on the basis of CD69 expression. It is important to provide additional evidence to confirm that these cells are actually Th2 TRM cells.

4. The authors established a novel in vitro model to study the differentiation and function of pTh2 cells. They employed bulk RNA-Seq profiling to demonstrate that the pTh2 cells differentiated in vitro reflect lung-derived pTh2 cells. It is important that these analyses are performed with single cell RNA-Seq, to demonstrate that the heterogeneity of the in vitro differentiated pTh2 cells reflects the lung derived pTh2 cells.

5. Extended Data Figure 1. Please show the data for BAL neutrophils.

Reviewer #2

(Remarks to the Author)

Khan et al. investigated the regulatory mechanisms and heterogeneity of lung pathogenic Th2 cells in mice. Using single-cell RNA-sequencing and flow cytometry, the authors revealed the heterogeneity within interleukin-33 receptor ST2-expressing Th cells and identified the cell surface markers that can distinguish Th2-like regulatory T cells from the pathogenic Th2 (pTh2) subset. The authors established an in vitro model that induced pTh2 cells, which closely resembled the pTh2 cell characteristics observed in vivo, using GITR and TSLP stimulation. Furthermore, they underscored the crucial role of HDAC1 in regulating the pathogenic Th2 responses.

The identification of signature genes and surface markers for the pTh2 subset could significantly contribute to future research on pTh2 cells in vivo. Overall, the authors have thoroughly conducted experiments using a multimodal approach, and have clearly conveyed their findings in the manuscript. However, some refinements still need to be made before this study can be published. I have listed the following comments:

Major comments

1. As shown in Figures 3h, 3k, and 4f, the flow cytometry plots appear to be sparse. The authors should consider changing the data visualization method to contour plots or increasing the cell number for these analyses.
2. As shown in Figures 6f and 6g, the Gata3 expression showed an increasing trend, especially in the TSLP+DTA-1-stimulated condition. The authors should discuss the relationship between HDAC1 and Gata3, and describe the predicted mechanisms through which HDAC1 represses the effector functions.
3. Based on the violin plots in Figures 4c, the authors focused on Il5 and Tnfsf11; however, the differences in the expression levels between the groups, especially in Il5, were not readily apparent. Adding box plots and p-values to each violin plot may help to highlight these differences.

Reviewer #3

(Remarks to the Author)

In their manuscript, the authors have used remarkable and state of the art technologies to follow their hypothesis. They have combined single cell RNA-seq with bulk RNA-seq and animal experiments to distinguish their peTh2 cells from tmTh2. Furthermore, they have identified multiple T-cell clusters including an interesting Treg/Th2 cluster. Looking further into possible mechanisms they have reported some work on HDAC1, MAP kinase and AP1.

I have the following comments:

Major Comments

- Eosinophils in Ext. Figure 1 are not convincing. CD11b is a marker for Eos. I would like to see a much more detailed gating strategy.
- In their HDM model, where are neutrophils? If it is mixed phenotype more neutrophils should be shown, which is expected from an HDM model. If it is a Th2 phenotype, 20% Eos in very weak for the WT mice
- Can the authors show real AHR that the mice are getting sick?
- I would require to see FMOs or isotype controls for the FACS plots, especially cytokines.
- The distinction of the Th2 cells is not very convincing.
- Treg/Th2 cells are they destabilized Treg cells? The authors should do cell tracing experiments.
- Tregs destabilized toward Th2 are very pathogenic. It was shown by other researchers; can the authors comment on that?
- As I am not a bioinformatician, I would require RNAseq and scRNA seq data in CSV files to check viability of the genes.
- Previous studies have shown that Th2 like Treg cells lack AREG. The analysis in cluster 6 is very worthy to be re-investigated or much more detailed discussion should be invested in AREG. Rudensky and Chatila groups have shown AREG-Treg cells connection differently.
- Chatila et al. defined Notch4 in destabilized Treg cells toward Th2 cells as a marker. Can the authors define that marker in their Treg/Th2 cluster, as they have shown RBPj as well?
- Did the authors check group 2 and 3 for their IL-13 production? If yes, can they comment on it?
- Fig. 4c, AREG seems significantly different between the groups. Can the authors discuss more about it?
- The Role of HDAC1 in explaining the mechanism of action in peTh2 is lacking. This is a major caveat of the paper that needs addressing.
- Can the authors explain why the peTh2 cells are enriched with IL-10? Is the clustering correct?
- TSLPR was never mentioned in the analysis. It comes as a sudden. Was TSLPR found in the cells in the scRNA-seq? if it induces pathogenic features, it should be present in the transcriptomic analysis.
- Production of IL-9 and GM-CSF! Are the cells more into a Th9 transient cell state? Can the authors comment on these findings? Are peTh2 are more of a Th9 cells?
- The IL-5+ cells in the 3H3 group is comparable to GITR in HDAC1 cKo group. Furthermore, the number of mice used for the analysis is very low (3), more mice should be used to confirm these findings.

- The authors neglected ex-Treg cells entirely in their analysis and clustering. It is very important to understand how their Treg/th2 cells cluster compare to ex-Treg cells.
- The explanation of Fig 7b/c is not convincing on IL-5 and IL-13. In WT IL-13 is much higher than HDAC cKO. Can the authors review the results?
- Why wasn't IL-10 and AREG (Amphiregulin) shown in proteomics data? Can the authors explain?
- The final part on explaining the molecular mechanism is very shallow and weak. Epigenetic work on these cells need to take place to explain how HDAC1 control IL-5 and IL-13 loci without changing GATA3.
- The peTh2 cells are bad per authors definition, do they differ in their work in transfer exp with parasitic infection from regular Th2 and TmTh2? Will they not act or are they only super stimulated cells?

Minor Comments

- "Ref" by references 9,10,18)
- Authors should show FACS plots for KLRG1 and CD27 in Ext. Fig4
- Anti-GITR blocking antibodies is very confusing, please rephrase it.
- Can the author explain briefly how they normalized 2 different RNA-seq for their comparison in 7 d,e or was the comparison only on gene numbers?
- FACS plots should be unified in their shape between all figures main and extended.

Version 1:

Reviewer comments:

Reviewer #1

(Remarks to the Author)

The authors have addressed my comments. I have no further comments.

(Remarks on code availability)

Reviewer #3

(Remarks to the Author)

no more comments

(Remarks on code availability)

Point-by-point response to the reviewer's comments

Reviewer #1 (Remarks to the Author):

Summary of findings

In this novel and interesting study, the authors employed a mouse model of HDM exposure to characterize Th2-memory responses in the lung and dissect the role of HDAC1 in the regulation of HDM-induced airway inflammation. They found that HDAC1-cKO mice had exaggerated eosinophilic airway inflammation and enhanced IL-5 and IL-13 responses in ex vivo stimulated T-helper cells. They then employed IL-13 tdTomato reporter mice to isolate IL-13 positive T-helper cells, non-IL-13 producing T-helper cells, and naïve T-helper cells for single cell profiling. They showed that the T cell population was divided into 9 distinct clusters. The cells in cluster 2 expressed the highest levels of Th2 effector molecules (IL-4, IL-5, IL-13, Areg) and accordingly were designated pathogenic effector Th2 (peTh2). The cells in cluster 5 expressed moderate levels of Th2 effectors in combination with the TRM marker CD69, and accordingly were designated Th2 Trm cells. GSEA analysis demonstrated that the T-helper subset had shared overlapping gene signatures, and therefore the authors established a flow cytometry to better define the subsets. They showed that peTh2 cells can be identified as ST2+CD27-KLRG1-CD69^{low}PD1^{high}, and Th2 Trm cells as ST2+CD27-KLRG1-CD69^{high}PD1^{high}.

The authors then set out to establish an in vitro model to study the differentiation of pTh2 cells. Given that the pTh2 cells express GITR, the authors differentiated naïve CD4 T cells in Th2 promoting conditions, with or without TSLP and/or a GITR agonist. They found that the combination of TSLP and the GITR agonist resulted in the highest production of Th23 effector cytokines, and this was further augmented in the HDAC1-cKO cells. The authors also performed bulk RNA-Seq to demonstrate that the in vitro pTh2 cells had a similar molecular signature to lung derived pTh2 cells.

Finally, the authors found that the lung derived and in vitro pTh2 cells were enriched with components of the AP-1, MAPK, and NF-κB signaling pathways, and they employed the small molecule inhibitors to target AP-1 (T-5224) and p38 MAPK (SB 203580) on the in vitro differentiation of pTh2 cells. They found that SB 203580 markedly suppressed IL-5 and IL-13 responses but not IL-4 or gata-3 levels in WT and HDAC1-cKO pTh2 cells. In contrast, AP-1 inhibition with T-5224 had no effect on pTh2 differentiation. The authors conclude that their findings resolve the heterogeneity of murine lung pTh2 cell subsets and unveil their transcriptional and phenotypic profiles, identify HDAC1 as a master regulator of pTh2 cells, and establish a novel in vitro model to generate pTh2 cells in vitro.

Reply: We thank this reviewer for the positive evaluation of our work and for recognizing its novelty. We also are grateful for the comments which greatly improved our manuscript.

Major comments

1. The findings in this manuscript are largely based on a single cell profiling experiment consisting of 10 samples. The experiment does not contain any biological replicates. To ensure the reproducibility and robustness of the findings, it is important to replicate the experiment.

Reply: We agree with this reviewer that we used 10 samples for one experiment, consisting of 10 distinct cell populations labeled with different hashtags to distinguish them after single cell-RNA sequencing. However, the experiment was conducted twice on two separate days, and the analysis results from the integration of these two independent experiments. In addition, 3 animals were used for one sample to get enough cells for sequencing, and also to increase normalization of the data. This is to our knowledge a standard procedure for this method.

The experimental details were in the figure legend of Figure 1: *“Fig.1| scRNA-seq analysis of lung CD4+ T cells uncovers the heterogeneity of pTh2 cells in response to HDM. a, Experimental design for scRNA-seq analysis of lung CD4+ T cells. After obtaining single-cell suspensions from the lungs of WT (HDAC1^{fl/fl} x CD4-Cre^{-/-}) and HDAC1-cKO (HDAC1^{fl/fl} x CD4-Cre^{+/-}) IL-13 tdTomato-reporter mice that were sensitised and challenged with PBS or HDM (as in Extended Data Fig. 1a), we sorted the following lung CD4+ T cells: naïve (TCRb⁺CD4⁺CD62L⁺CD44⁻IL-13⁻), IL-13⁻ Th (TCRb⁺CD4⁺CD62L⁻CD44⁺IL-13⁻), and IL-13⁺ Th (TCRb⁺CD4⁺CD62L⁻CD44⁺IL-13⁺) cells. A total of ten samples based on the two genotypes (WT or HDAC1-cKO) and experimental conditions (HDM or PBS) were obtained. Each sample was labelled with a unique hashtag oligonucleotide (HTO). All ten samples were pooled for single-cell RNA-sequencing (scRNA-seq) analysis. **Two independent scRNA-seq experiments were integrated for the analyses.**”*

The material and method section also contained information on how the analysis was performed, and the two experiments merged in the analysis of scRNA-seq data section on page 14:

*“After processing each dataset separately, we merged them and performed normalisation, feature selection, data scaling, dimensionality reduction, and clustering, as previously mentioned. **We then integrated both datasets** by canonical correlation analysis (CCA) using the FindIntegrationAnchors and IntegrateData functions, then scaled the integrated data, performed dimensionality reduction, followed by clustering of the cells using FindNeighbors (on the first 15 principal components) and FindClusters (at a resolution of 0.5).”*

However, given the complexity of the experimental set-up, we thank this reviewer to have pointed out a potential unclarity. Therefore, we added the following sentence on page 13 in the material and method part at the end of the sample preparation section of the single-cell analysis of lung CD4⁺ T cells:

“The samples were then resuspended in 100 μ l PBS containing 0.04% BSA, and cells were counted. 3,000 cells were taken from each sample and pooled. A total of 30,000 cells (from 10 samples) were submitted for each scRNA-seq experiment.

We added the sentence below in the methods part of our manuscript on page 14 from line 655 to 657:

Two independent experiments with 10 samples each were performed and data integrated for the analysis. Each sample is a pool of cells from three different experimental animals.”

2. The single cell clusters were annotated using a series of known marker genes selected by the authors. It is important to confirm the cell annotations using an unbiased reference-based approach such as Azimuth.

Reply: We thank this reviewer for this comment and recommendation of an unbiased reference-based approach. We acknowledge that annotation of the different clusters is challenging and sometimes misleading, and that there is a crucial need for unbiased cluster annotations. The Azimuth web application is however not suitable for our data, as it relies on human immune cell annotations, which are different from murine immune cell annotations. We therefore used the SingleR package which can be linked to the state-of-the-art murine reference database in Immgen (**Revision Fig. 1**).

However, our pathogenic Th2 cells were identified as NKT cells and ILC. The annotation is not suitable for us, as it defines subsets in our UMAP which were not present in our cell preparation (such as NKT cells, ILC, and $\gamma\delta$ T cells). The reason for these incorrect annotations relies on the specialized T cell subtypes (T helper lineages) that are the focus of our work and which are not so far included in official and dedicated databases like Immgen. With our experimental approach, we “zoomed” into the T helper cell subtypes during allergic airway inflammation. Therefore, transcriptional signatures will be more similar on a whole when compared to other immune cell lineages, and a more refined signature is needed in our case, focusing specifically on T helper subsets and a broader immune cell classification including NKT cells and $\gamma\delta$ T cells as found in Immgen cannot be applied. We prefer to keep our manual annotation which was done in agreement with the annotation method used by Tibbit et al., 2019 (PMID: 31231035), taking into account master transcription factors, cytokine production and definition of cell subsets based on literature.

The limitations of automated cluster annotations are also described in PMID: 38187377, which acknowledge “a plethora of missing annotations”, giving Th2 cells as an example of a missing signature.

Revision Fig.1| Cluster annotation by SingleR. a, Cluster annotation by SingleR. **b**, UMAP plots of $CD4^+$ T cells colour-coded based on selected marker genes for NK cells, ILC, and T cells.

3. The Th2-tissue resident memory cells were defined primarily on the basis of CD69 expression. It is important to provide additional evidence to confirm that these cells are actually Th2 TRM cells.

Reply: We agree that we defined the Th2 TRM primarily on the basis of CD69 expression along with ST2 expression. CD69 was confirmed as a marker for lung resident memory CD4⁺ T cells in a house dust mite allergy model as early as 5 days after sensitization in the seminal study performed by the group of Marion Pepper in 2016 (PMID: 26750312). Further, CD69 has recently been shown as a Th2 TRM marker in an ovalbumin plus *Alternaria* allergy model by the group of Hirohito Kita (PMID: 36720287). Further characterization of these TRM led to the conclusion that they are CCR7 and CD62L negative and ST2 positive. CD4⁺ TRM were further shown by the group of Donna Farber to express high levels of CD11a (PMID: 22058417). We can confirm that our Th2 TRM in cluster 5 express high levels of *Il1r1* and *Itgal* (encoding CD11a) but are negative for *Ccr7* and *Sell*. In addition, TRMs were shown to be *S1pr1* negative and *Klf2* low (PMID: 38381299) which is in agreement with the expression pattern of cells in cluster 5. We can therefore summarize that the cells in cluster 5 have a TRM specific signature, being not only Cd69 positive, but also *Il1r1* and *Itgal* positive, and *Ccr7*, *Sell*, *S1pr1* and *Klf2* negative or low, corresponding to a core transcriptional signature for TRM (PMID: 28930685). In contrast to human Lung CD4⁺ TRM cells which express CD103 as tissue residency marker, mouse lung CD4⁺ TRM cells were shown to be CD103 negative and mainly defined by CD69 expression (PMID: 36032142). In agreement with published studies, cells in cluster 5 are Cd103 negative.

We agree that in an inflammatory chronic condition like allergic airway disease, continuous stimulation of the cells might occur. CD69 is also a marker of recent activation, which might lead to wrong conclusions. Therefore, we assessed the expression of another activation marker like CD25 in cluster 5 (a similar reasoning and analysis can be found in a study from the group of Donna Farber, PMID: 28930685). In contrast to *Cd69*, *Cd25* (*Il2ra*) was expressed at very low level in cluster 5, only minimally higher compared to naïve CD4⁺ T cells, and much lower compared to the effector cells of cluster 2, which are still actively expressing cytokine mRNA and correspond to a more recently activated cell type. In addition, recently activated cells would show proliferating features like expression of *Mik67*, which cells in cluster 5 don't express.

A summary of these markers is included in **Extended Data Fig. 2 and Fig. 4e**.

In light of this deeper analysis of our scRNA-seq data and the current literature, we conclude that cells contained in cluster 5 have strong features of Th2 TRM cells and we decided not to perform additional *in vivo* experiments like the intravenous injection of fluorescent anti-CD45 antibody to mark tissue resident cells specifically.

However, to fully understand the nature and regulation of these Th2 TRM cells, and the involvement of HDAC1 in the control of tissue resident memory, a project dedicated to this aspect will be started in the lab, including tissue specific labeling of these cells.

Extended Data Fig. 2 | Selected markers, transcriptional regulators, and mediators. a-d, scRNA-seq analysis of lung CD4⁺ T cells in response to PBS or HDM. **a**, UMAP showing all fourteen lung CD4⁺ T cell clusters identified. **b-d**, Defining the identity of the clusters in **a**. Dot plots showing the expression of selected markers (**b**), transcriptional regulators (**c**), and Th-related mediators and transcription factors. **TRM signature genes used in our reply are highlighted.**

e

Fig.4e | Violin plots of marker genes associated with tissue residency.

4. The authors established a novel *in vitro* model to study the differentiation and function of pTh2 cells. They employed bulk RNA-Seq profiling to demonstrate that the pTh2 cells differentiated *in vitro* reflect lung-derived pTh2 cells. It is important that these analyses are performed with single cell RNA-Seq, to demonstrate that the heterogeneity of the *in vitro* differentiated pTh2 cells reflects the lung derived pTh2 cells.

Reply: We agree with this reviewer that we designed a novel *in vitro* model to study the differentiation and function of pTh2 cells. As *in vitro* cultures are related to recently activated cells, the cells would be expected to rather correspond to effector type cells *in vivo*. This is indeed what we observed, as shown in **Fig. 7d** and **Extended Data Fig. 6d** by GSEA analysis, and summarized here in **Revision Fig. 2** for a better overview:

Revision Fig.2| a. Enrichment plot showing a comparison of WT pTh2 cells and WT Th2 cells to lung peTh2 cells. **b.** Enrichment plot showing a comparison of WT pTh2 cells and WT Th2 cells to lung Th2 Trm cells.

7d) than that of the Th2 TRM signature genes (1.04) (Extended Data Fig. 6d), implying a strong resemblance of the *in vitro* cultured pTh2 cells with the *in vivo* peTh2 cells, but not the Th2 TRM cells. This is further supported by the extremely low false discovery rate of 4.30E-08 for peTh2 signature genes compared to 0.37 for Th2 TRM signature genes. These data underscore the strong similarity of cultured pTh2 cells with peTh2 *in vivo* cells.

These results were explained in the result section on page 8 line 369 to 373: “We further compared the *in vitro* pTh2 cells with lung Th2 Trm cells. Although the *in vitro* pTh2 cells shared some features with lung Th2 Trm cells, their similarity was less pronounced (Extended Data Fig. 6d-f and Supplementary Table 8) as compared with the lung peTh2 cells (Fig. 7d-f). This is expected as the Th2 Trm cells already acquired tissue residency features with diminished effector program (Fig. 4d, e).”

We also highlighted this in the discussion on page 11 line 502 to 506: “Our data uncover that co-stimulation of GITR together with TSLPR, induces pTh2 differentiation *in vitro*. We confirm that these *in vitro* generated pTh2 cells are highly similar to lung peTh2 cells and exhibit enhanced upregulation of pathogenic molecules and TFs. Therefore, our work unravels the importance of co-stimulatory factors like ligands of TNFRSF members together with TSLP to drive the differentiation of pTh2 cells”.

To further confirm a strong association of *in vitro* cultured pTh2 cells with *in vivo* generated peTh2 cells and not Th2 TRM, we performed immunophenotyping of our cultured pTh2 cells. We cultured the cells for 2 or 5 days. On day 2, the cells still retain signs of activation like CD69 and also express CCR7. On day 5, the pTh2 cells showed diminished expression of both CCR7 and CD69, indicating that they

have acquired an effector phenotype but do not have a tissue retention property like Th2 Trm cells (Revision Fig. 3).

Revision Fig.3| Assessment of tissue residency markers in *in vitro* cultured Th2 and pTh2. Naive CD4⁺ T cells from WT mice were cultured under Th2 and pTh2 conditions for 2 or 5 days. CCR7, CD69 and PD-1 were assessed by flow cytometry.

5. Extended Data Fig. 1. Please show the data for BAL neutrophils.

Reply: Neutrophils are hardly induced in our HDM model (Revision Fig. 4c and 3d). A possible reason for this could be the low level of endotoxin in our house dust mite extract. Such phenomenon has been reported previously (PMID: 31369802).

Revision Fig.4| Cellular BAL composition of healthy and diseased WT and HDAC1-cKO mice. a, Eosinophils were identified as Siglec F⁺ and CD11c⁻ by flow cytometry. **b,** Expression of CD11b by eosinophils. **c,** Neutrophils (Ly6G⁺ and CD11b⁺) in BAL of mice.

Reviewer #2 (Remarks to the Author):

Khan et al. investigated the regulatory mechanisms and heterogeneity of lung pathogenic Th2 cells in mice. Using single-cell RNA-sequencing and flow cytometry, the authors revealed the heterogeneity within interleukin-33 receptor ST2-expressing Th cells and identified the cell surface markers that can distinguish Th2-like regulatory T cells from the pathogenic Th2 (pTh2) subset. The authors established an in vitro model that induced pTh2 cells, which closely resembled the pTh2 cell characteristics observed in vivo, using GITR and TSLP stimulation. Furthermore, they underscored the crucial role of HDAC1 in regulating the pathogenic Th2 responses.

The identification of signature genes and surface markers for the pTh2 subset could significantly contribute to future research on pTh2 cells in vivo. Overall, the authors have thoroughly conducted experiments using a multimodal approach, and have clearly conveyed their findings in the manuscript. However, some refinements still need to be made before this study can be published. I have listed the following comments:

Reply: We thank this reviewer for the detailed and positive assessment of our work and the comments which greatly improved our manuscript.

Major comments

1. As shown in Figures 3h, 3k, and 4f, the flow cytometry plots appear to be sparse. The authors should consider changing the data visualization method to contour plots or increasing the cell number for these analyses.

Reply: We thank this reviewer for this recommendation. These figures represent rare cell populations, and we acquired already one million cells in our second and third experiments. However, we changed the data visualization method to contour plots and changed the figures accordingly in **Revision Fig.5:**

Fig. 3h-l

Fig. 4f

Revision Fig.5 Improvement of data visualization

2. As shown in Figures 6f and 6g, the Gata3 expression showed an increasing trend, especially in the TSLP+DTA-1-stimulated condition. The authors should discuss the relationship between HDAC1 and Gata3, and describe the predicted mechanisms through which HDAC1 represses the effector functions.

Reply: The reviewer is right, the GATA3 positive cells showed an increasing trend in the pTh2 cells despite no difference between the WT and HDAC1-cKO cells. We performed ATAC-seq of WT and HDAC1-cKO Th2 and pTh2 cells cultured for 24 and 48 hours after activation to better understand the possible relationship between HDAC1 and GATA3 for regulation of pTh2, as well as the possible mechanism of HDAC1-mediated repression of effector cytokines. We have added a totally new figure (**Fig. 9**) in the manuscript and propose a predicted mechanism in **Fig. 9f**.

3. Based on the violin plots in Figures 4c, the authors focused on Il5 and Tnfsf11; however, the differences in the expression levels between the groups, especially in Il5, were not readily apparent. Adding box plots and p-values to each violin plot may help to highlight these differences.

Reply: We agree with the reviewer that the box plot representation and p-values for each violin plot allow a more accurate analysis of our data. We modified the figures accordingly:

Fig.4| Loss of HDAC1 augments the pathogenicity of pTh2 subsets and Th2 Trm cell generation. c,d, Violin plots of selected pathogenic marker genes in peTh2 cells (c) and Th2 Trm cells (d) from WT and HDAC1-cKO mice.

Reviewer #3 (Remarks to the Author):

In their manuscript, the authors have used remarkable and state of the art technologies to follow their hypothesis. They have combined single cell RNA-seq with bulk RNA-seq and animal experiments to distinguish their peTh2 cells from tmTh2. Furthermore, they have identified multiple T-cell clusters including an interesting Treg/Th2 cluster. Looking further into possible mechanisms they have reported some work on HDAC1, MAP kinase and AP1.

Reply: We thank this reviewer for acknowledging the effort and technologies used in our work, and for providing helpful comments which led to a strong improvement of our work and to an insightful scientific discussion.

I have the following comments:

Major Comments

- Eosinophils in Ext. Figure 1 are not convincing. CD11b is a marker for Eos. I would like to see a much more detailed gating strategy.

Reply: We used the gating strategy recommended by the bio-protocol of Hongwei Han and Steven F. Ziegler (doi: [10.21769/bioprotoc.859](https://doi.org/10.21769/bioprotoc.859)). Indeed, all the eosinophils in the HDM exposed mice expressed CD11b (Revision Fig. 4b). However, SiglecF⁺ and CD11c⁻ have been widely used as markers for identifying eosinophils (PMID: 26414117; PMID: 32579670; PMID: 30250029).

Revision Fig.4| Cellular BAL composition of healthy and diseased WT and HDAC1-cKO mice. a, Eosinophils were identified as Siglec F⁺ and CD11c⁻ by flow cytometry. **b,** Expression of CD11b by eosinophils. **c,** Neutrophils (Ly6G⁺ and CD11b⁺) in BAL of mice. **d,** Graph shows the frequency of neutrophils in c.

- In their HDM model, where is neutrophils? If it is mixed phenotype more neutrophils should be shown, which is expected from an HDM model. If it is a Th2 phenotype, 20% Eos in very weak for the WT mice

Reply: We agree with this reviewer that in many HDM models, neutrophils are induced. However, the magnitude of the induction varies, owing to the heterogeneity of the HDM extracts (PMID: 31369802). An example of low neutrophil counts in BAL can be found in literature (Figure 1; PMID: 28798029). It was shown that recruitment of neutrophils is highly dependent on the HDM extract, as some contain higher amounts of endotoxin. Our HDM extract is almost endotoxin free and doesn't induce a strong recruitment of neutrophils. This is in agreement with results with endotoxin low HDM extracts (PMID: 31369802).

In addition, dependent on the HDM lot a range of 5 to 40 % eosinophil recruitment can be expected (PMID: 31369802). We have on average 1×10^5 eosinophils in the BAL of diseased mice, which is in agreement with data from the group of Talal Chatila (Extended Data Fig. 6, PMID: 32929274).

- Can the authors show real AHR that the mice are getting sick?

Reply: We performed one round of airway resistance measurement and could show that the WT mice exposed to HDM had enhanced AHR (**Revision Fig. 6**), indicating that the mice are sick.

Revision Fig. 6 | Airway hyperresponsiveness in healthy and diseased WT C57Bl/6 mice. Mice were sensitised and challenged with PBS (healthy) or HDM (diseased) as shown in Extended date Fig.1a. On day 16, airway hyperresponsiveness (AHR) was assessed by exposing the mice to increasing concentrations of methacholine. The left panel shows all the concentrations of methacholine used. The right panel shows the lung resistance of the mice in response to 50 mg/ml of methacholine. RI: Lung resistance.

- I would require to see FMOs or Isotype controls for the FACS plots, especially cytokines.

Reply: We usually use negative populations as control. We use the PBS control group to show that cytokines are not produced in absence of disease as shown in **Extended Data Fig. 1f**.

Extended Data Fig.1| HDAC1 is essential to restrict HDM-induced airway inflammation. *Ex vivo* restimulation of lung cells from WT and HDAC1-cKO control and diseased mice. Lung cells were restimulated with PMA and ionomycin in the presence of GolgiStop and GolgiPlug for 4 hours followed by cytokine analyses by flow cytometry. **f**, Representative flow cytometry plots showing IL-5 and IL-13 expression in lung Th cells (gated on TCR β^+ CD4 $^+$ CD44 $^+$).

We provide here a figure (**Revision Fig. 7**) showing our differentiation of T helper cells *in vitro*, and using FMO for IL-5 as an example. The figure shows that the IL-5 antibody is specific to IL-5 as the cytokine is unstained in the FMO cells.

Revision Fig. 7| FMO of Th2 and pTh2 cells. WT and HDAC1-cKO naïve CD4 $^+$ T cells were cultured under Th2 and pTh2 conditions for 5 days, and intracellular IL-5 and IL-13 was measured by flow cytometry (top panels). The bottom panels show FMO for IL-5.

- The distinction of the Th2 cells is not very convincing.

Reply: For scRNA-seq, lung cells from healthy and diseased animals were homogenized and IL-13 positive CD4⁺ T cells (detected via the IL-13 reporter) sorted. We were able to identify two clusters (cluster 2 and cluster 5) consisting of *Il13*, *Il5*, *Il4*, *Gata3* positive cells, which are to our knowledge accepted markers for Th2 cells. In addition, cells in these two clusters originated from the sorted IL-13 positive cells. However, a key difference between the two clusters is: cells in cluster 2 have a transcriptional signature corresponding to Th2 cell with active effector program whereas cells in cluster 5 have less type 2 cytokine transcripts but with enhanced expression of *Gata3* and *CD69*. Both clusters are Foxp3 negative. To the best of our knowledge, these features define these cells as Th2 cells and are distinct from the other T helper subsets.

- Treg/Th2 cells are they destabilized Treg cells? The authors should do cell tracing experiments.

Reply: We thank this reviewer for this insightful question and the interesting suggestion. The Treg/Th2 cells in cluster 6 express high levels of Foxp3 but very low Th2 cytokine transcripts (like *Il4*, *Il5* or *Il13*). We would however expect destabilized Tregs to produce Th2 cytokines as shown by the group of Talal Chatila (PMID: 25769611). Extensive trajectories of tissue adaptation of Tregs were performed by the group of Sarah Teichmann (PMID: 30737144), and showed expression of *Rora*, *Gata3* and *Il1r1* as hallmarks of tissue adaptation (**Extended Data Figure 2b,c; Supplementary Table 2**). We could also detect these genes in our Treg/Th2 cells from cluster 6, suggesting that these cells are tissue adapted Treg cells. In addition, the majority of the cells in cluster 6 have a high expression of *Ikzf2* (Helios) (**Extended Data Figure 2c, Supplementary Table 2**) which is linked to Treg stability (PMID: 32929274, PMID: 30620397, PMID: 34561855, PMID: 32929274). These observations together suggest that the majority of the cells in cluster 6 are stable tissue adapted Tregs rather than destabilized.

Intriguingly, approximately 35% of cells in cluster 6 originated from IL-13 reporter positive samples from mice exposed to HDM, as shown in **Revision Fig. 8**. Therefore, we can assume that some cells in this cluster produced IL-13. It was shown that IL-13 produced by ST2-positive Tregs is protective by helping to limit tissue inflammation after lung injury (PMID: 30779711). We might also have some Tregs in cluster 6 which secrete IL-13. Whether these cells are protective or harmful remains to be assessed.

Revision Fig.8| Distribution of sorted cell subsets in cluster 6.

We agree that experiments using a lineage tracing approach based on a *Rosa26* Stop-flox YFP reporter crossed with a *Foxp3*-specific Cre recombinase fused to an enhanced green fluorescent protein (EGFP) as used by the group of Talal Chatila (PMID: 25769611) to trace ex Treg cells would be of high interest. We plan to perform such experiments, in addition to IL-13 f/f mice as well as functional assays to fully understand the nature, heterogeneity and functionality of the Treg cells in cluster 6. We consider however that this is a new project, which is extremely exciting, but is beyond the time frame of these revision experiments. We will ask for additional funding to perform these experiments and thank this reviewer for opening these new perspectives.

However, we acknowledge that this point is important and needs to be highlighted more. Therefore, we added these sentences in the result on page 5 from line 183 to line 189:

“During allergic responses in the lung, Tregs might get subverted and lose their regulatory functions, which licenses them to acquire a Th2 signature and promote allergic airway inflammation⁴⁷. A hallmark of Treg subversion is the expression of the Notch4 receptor⁴⁷. The here identified Treg/Th2 cells do not express Notch4, are Foxp3 positive (Supplementary Table 2) and show high expression of *Rora*, *Ikzf2*, and *Il2ra* (Fig. 1c and Supplementary Table 2) suggesting that these cells are stable, tissue adapted Tregs with repressive ability as previously described^{47, 48, 49, 50}. However, to fully dissect the biology of Treg/Th2 cells, lineage tracing experiments are warranted”.

Collectively, our data resolve the heterogeneity of murine lung pTh2 cells, reveal their similarities with previously overlooked lung Il1rl1-expressing Th cells (Treg/Th2), and highlight the need to define markers to distinguish the pTh2 cell subsets from the Treg/Th2 subset”.

And these sentences in the result section on page 10 from line 457 to line 460:

*“In addition, we identify a unique Th subset that has features of both Treg and Th2 cells (Treg/Th2). Cells in this cluster express *Rora*, *Gata3* and *Il1r1* (encoding ST2), which were identified as markers of tissue adaptation for Tregs⁴⁸. These cells also express high levels of *Ikzf2* which is linked to Treg stability^{47, 49}. These observations suggest that the majority of the Treg/Th2 cells are stable tissue adapted Tregs. Intriguingly, the pTh2 cell subsets show some similarities with the Treg/Th2 cell subset. Many marker genes previously attributed to lung pTh2 cells such as *Il1r1* (ST2) and *Bhlhe40* are not unique to the pTh2 cell subsets”.*

- Tregs destabilized toward Th2 are very pathogenic. It was shown by other researchers; can the authors comment on that?

Reply: We agree with this reviewer that Tregs destabilized toward Th2 were described as very pathogenic as shown by the group of Talal Chatila (PMID: 25769611) and Mark Wilson (PMID: 28507062). The role of Tregs with Th2 features is still under debate and merits further investigation. We will study the role of these cells and their functionality in a follow-up project as mentioned in our reply to the previous question; however, we consider that this very intriguing and interesting aspect merits more time and study and is beyond the time frame of this current revision experiments. We therefore rather concentrated our effort here on the Th2 effector cells, and on answering the mechanistic part and epigenetic regulation of pathogenic Th2 cells by HDAC1.

- As I am not a bioinformational, I would require RNAseq and scRNA seq data in CSV files to check viability of the genes.

Reply: We provided these files as supplementary tables and were included in the initial submission. This reviewer might not have received these files. Below is a link to access our manuscript on Bioarchive, and you can access all the tables:

<https://www.biorxiv.org/content/10.1101/2023.08.10.552772v1.supplementary-material>

- Previous studies have shown that Th2 like Treg cells lack AREG. The analysis in cluster 6 is very worthy to reinvestigated or much more detailed discussion should be invested in AREG. Rudensky and Chatila groups have shown AREG-Treg cells connection differently.

Reply: We agree that amphiregulin produced by Tregs is an important factor in tissue repair as shown by the group of Alexander Rudensky (PMID: 26317471) after acute lung damage upon influenza virus infection. In addition, the group of Talal Chatila (PMID: 32929274) showed that Tregs get subverted during allergic airway inflammation and become pathogenic due to a Notch4-GDF15 signaling in Tregs. Amphiregulin was however not shown in this study. In a further study by the same group (PMID: 33915108), Notch4 signaling was shown to restrain tissue repair by Treg by inhibiting amphiregulin in a lung viral disease setting. To our knowledge, there are no data shown in publications revealing that Th2 like Treg secrete less or even lack Amphiregulin in an allergy setting. It was however shown by the group of Toshinori Yakahama (PMID: 29958800) that amphiregulin produced by pathogenic memory Th2 cells induces airway fibrosis via osteopontin produced by eosinophils. These data are in agreement with our data showing that pTh2 express Areg. In line with our data, as GATA3 induces AREG expression (PMID: 24631153), we can assume that Th2 cells and Th2 like cells expressing GATA3 can produce amphiregulin. As we might not have subverted Tregs in our disease model, these Tregs might still produce amphiregulin. An interesting question which to my knowledge is still open, is to understand whether amphiregulin produced by Tregs is protective in contrast to amphiregulin produced by pTh2 which induces fibrosis, possibly through different cellular localizations and interactions. This is definitively a new aspect that we want to include in our future studies. Together with the absence of Notch4 expression by the Tregs in cluster 6 of our study, we might conclude that we don't have pathogenic subverted Tregs in our *in vivo* model, and that we describe and characterize all pathogenic Th2 cells in our study as mentioned in the title of the manuscript. A second project focusing on Tregs is definitively very important in the future.

We have added however a section in the result part on page 5 from line 175 to line 182 to address this important point raised by this reviewer:

“Similar to pTh2 cells, Treg/Th2 cells also expressed *Areg* and *Il10*, encoding amphiregulin (Areg) and IL-10, respectively (Fig. 1c). Both Areg and IL-10 are regarded as anti-inflammatory, tissue-protective and functional “Treg”-associated mediators^{44, 45}. However, recent reports indicated context-dependent roles for these mediators. For example, Areg was shown to play a protective role during viral infection by inducing tissue repair⁴⁴. However, Areg secretion by pTh2 cells in response to HDM potentiates lung fibrosis⁴³. Also, IL-10 production by HDM-specific T cells augments Th2 cell differentiation and allergic asthma⁴⁶. Therefore, the protective effect of Areg and IL-10 might be dependent on cell type, tissue localization, and cellular interactions”.

- Chatila et al. defined Notch4 in destabilized Treg cells toward Th2 cells as a marker. Can the authors define that marker in their Treg/Th2 cluster, as they have shown RBPj as well?

Reply: We thank this reviewer for this nice suggestion. We don't find Notch4 expression in our pathogenic effector Th2, Th2 Trm and Treg/Th2 cells. However, we observe RBPj expression in pathogenic effector Th2, Th2 Trm and Treg/Th2, which might not be related to Notch4 expression (**Revision Fig. 9**). We don't have an explanation yet for the enhanced expression of RBPj in absence of Notch4 expression in Treg/Th2 cells, and also why the expression of RBPj is higher in pathogenic effector Th2 and Th2 Trm compared to Treg/Th2 cells. Expression of Rbpj was also observed in lung pTh2 cells (PMID: 31231035). It has been described that RBPj might function in Notch-dependent and Notch-independent manner (PMID: 23651858; PMID: 35848919). For now, we only can speculate that RBPj might act on cells in cluster 2, 5, and 6 in a Notch-independent manner.

However, as Notch4 is an accepted marker of subverted Tregs, we might postulate that the Treg/Th2 cells are not subverted Tregs as they are lacking this receptor. One possible explanation why Treg/Th2 cells in our HDM induced allergy model don't get subverted might rely on the low endotoxin content of the HDM, and low recruitment of neutrophils, which in turn indicates low Th17 induction and IL-6 activity. As IL-6 was shown to be essential in driving Notch4 expression, this might be the underlying reason.

Revision Fig.9| Expression of transcription factors regulating Tregs and subverted Tregs in our scRNA-seq datasets.

- Did the authors check group 2 and 3 for their IL-13 production? If yes, can they comment on it?

Reply: If we understand this question correctly, this reviewer is asking for the IL-13 production of cluster 2 and cluster 3 from our single cell RNA-seq analysis, referring to pTh2 and activated cells respectively. Cells in cluster 2 had the highest Il13 expression, whereas cells in cluster 3 had very low Il13 expression. We didn't isolate these cells and re-stimulated them *in vitro* for cytokine production, as we would have to define exact markers for this purpose.

- Fig. 4c, AREG seems significantly different between the groups. Can the authors discuss more about it?

Reply: Yes, the expression of Areg is higher in WT pTh2 cells as compared to the HDAC1-cKO pTh2 cells. This indicates that its expression is independent of HDAC1. Cells in cluster 5 corresponding to Th2 TRM might have already downregulated the transcript and would need to be re-stimulated to induce AREG.

- The Role of HDAC1 in explaining the mechanism of action in pTh2 is lacking. This is a major caveat of the paper that needs addressing.

Reply: We agree with this reviewer that this is a major caveat and was missing. Therefore, we performed ATAC-seq of WT and HDAC1-cKO Th2 and pTh2 cells cultured for 24 and 48 hours after activation and obtained new insights into the possible mechanisms regulating the differentiation of pTh2 cells. We have now added a new figure (**Fig. 9**) to address this issue and provide a mechanism in **Fig 9f**.

Fig.9 | Type 2 cytokine expression is differentially regulated at chromatin level in pathogenic versus classical Th2 cells.

Editorial note: Panel f Created in BioRender. Boucheron, N. (2025) <https://BioRender.com/x77f144>

- Can the authors explain why the pTh2 cells are enriched with IL-10? Is the clustering correct?

Reply: Yes, our clustering is correct and was performed in accordance with the standard Seurat pipeline. However, we understand the surprise of this reviewer related to the production of IL-10 by pTh2 cells. Coming from a helper T cell expertise and performing CD4⁺ T cell cultures for over 20 years, I am used to detecting high levels of IL-10 in Th2 cultures. This is in agreement with the first studies discovering IL-10 (also known as: cytokine synthesis inhibitory factor) in Th2 cells (PMID: 2531194) and reviews on Th2 secreted cytokines (PMID: 10719286). We also detected IL-10 in our Th2 and pTh2 cultures (**Revision Fig. 10**), which is in agreement with the pTh2 cell transcriptional signature in our *in vivo* HDM allergy model.

Revision Fig.10| IL-10 secretion by Th2 and pTh2 cells. Naïve CD4⁺ T cells were isolated from WT and HDAC1-cKO mice, and cultured for 5 days under Th2 and pTh2 condition. IL-5 and IL-10 were measured by flow cytometry.

However, we agree that this point merits further consideration in the manuscript, as IL-10 is traditionally considered as an anti-inflammatory cytokine and as a prognostic marker for disease improvement during allergy immunotherapy (PMID: 29221580). It is shown to be mainly produced by Tregs and regulatory B cells *in vivo* (PMID: 31711770, PMID: 37837939). Its protective function in allergic responses is not completely clear, as exogenous administration of IL-10 to Balb/C mice in an ovalbumin-induced allergic airway model resulted in reduced airway eosinophilia (PMID: 7769104). Further IL-10 gene knock-out attenuated allergen-induced airway hyperresponsiveness in C57BL/6 mice (PMID: 11159016). However, IL-10 has been shown to be important in inducing Th2 cell-mediated allergic airway inflammation (PMID: 10671193, PMID: 32399548, PMID: 39394532). These discrepancies might be due to the specific effects of the cellular source of IL-10, the mouse background and the disease or inflammatory context (PMID: 38497670).

We added the section below in the result part of our manuscript on page 5 from line 175 to 182:

“Similar to pTh2 cells, Treg/Th2 cells also expressed *Areg* and *Il10*, encoding amphiregulin (Areg) and IL-10, respectively (Fig. 1c). Both Areg and IL-10 are regarded as anti-inflammatory, tissue-protective and functional “Treg”-associated mediators^{44, 45}. However, recent reports indicated context-dependent roles for these mediators. For example, Areg was shown to play a protective role during viral infection by inducing tissue repair⁴⁴. However, Areg secretion by pTh2 cells in response to HDM potentiates lung fibrosis⁴³. Also, IL-10 production by HDM-specific T cells augments Th2 cell differentiation and allergic asthma⁴⁶. Therefore, the protective effect of Areg and IL-10 might be dependent on cell type, tissue localization, and cellular interactions”.

- TSLPR was never mentioned in the analysis. It comes as a sudden. Was TSLPR found in the cells in the scRNA-seq? if it induces pathogenic features, it should be present in the transcriptomic analysis.

Reply: Yes, the reviewer is right. The TSLP receptor (TSLPR; encoded by *Crlf2*) is expressed by the pathogenic Th2 cells (**Extended Data Fig. 2 and Supplementary table 2**). Since our pathway analysis revealed an enrichment of IL-2/STAT5-regulated transcriptional program in the peTh2 cells, and that previous studies revealed that TSLP signaling via the TSLPR promotes Th2 cell differentiation and cytokine production in a STAT5-dependent manner (PMID: 21484783; PMID: 29535264), we reasoned that it might be an early signal for driving pTh2 differentiation. This has already been highlighted in the manuscript.

- Production of IL-9 and GM-CSF! Are the cells more into a Th9 transient cell state? Can the authors comment on these finding? Are peTh2 are more of a Th9 cells?

Reply: IL-9 and GM-CSF are also cytokines produced by Th2 cells (PMID: 10719286; PMID: 24332592; PMID: 28768806; PMID: 37146132). We acknowledge that the discrimination between Th9 and pathogenic Th2 is often not clear (PMID: 30658968). However, given the high production of Th2 cytokines, the cells in this study are rather defined as Th2 cells that produce IL-9.

- The IL-5+ cells in the 3H3 group is comparable to GITR in HADAC cKo group. Furthermore, the number of mice used for the analysis is very low (3), more mice should be used to confirm these finding.

Reply: We have increased the number of mice from 3 to 5, and as rightly pointed out by the reviewer, both DTA-1 (anti-GITR antibody) and 3H3 (anti-4-1BB antibody) have comparable effect in inducing pTh2 cytokines in absence of HDAC1 (**Extended Data Fig. 5e**).

Extended Data Fig.5| Flow cytometric analysis of pathogenic mediators and comparison of TNFRSF members. d,e, Comparing the impact of TNFRSF members in inducing a pathogenic program in Th2 cells. We isolated naïve CD4⁺ T cells (TCRβ⁺CD4⁺CD62L⁺CD44⁻) from WT and HDAC1-cKO mice and activated them with anti-CD3 and anti-CD28 in the presence of Th2-promoting conditions (IL-4, IL-2, anti-IFN-γ, and anti-TGF-β; collectively termed IL-4), or IL-4+TSLP alone, or IL-4+TSLP+DTA-1, or IL-4+TSLP+OX86, or IL-4+TSLP+3H3. We cultured the cells for 5 days and restimulated them with PMA and ionomycin in the presence of GolgiStop and GolgiPlug for 4 hours before cytokine analyses by flow cytometry. **d,** Graphs showing the frequencies of pathogenic Th2 mediators and GATA3 in WT cells (blue). **e,** Graphs showing the frequencies of pathogenic Th2 mediators and GATA3 in HDAC1-cKO cells (red).

- The authors neglected ex-Treg cells entirely in their analysis and clustering. It is very important to understand how their Treg/Th2 cells cluster compare to ex-Treg cells.

Reply: If I understand this reviewer correctly, ex-Treg cells would be Treg cells that lost expression of FoxP3. This aspect is of course of interest, but was not the focus of this study, which was on the differentiation of pathogenic Th2 cells. Cells in the Treg/Th2 cluster do not express Notch4, implying that they are not subverted Tregs. We are planning a follow-up study with fate mapping mice as specified earlier in reply to subverted Tregs to clearly understand the differentiation and function of Tregs during our allergic airway disease.

We added a sentence in the result part at page 5 line 175 to 182:

“During allergic responses in the lung, Tregs might get subverted and lose their regulatory functions, which licenses them to acquire a Th2 signature and promote allergic airway inflammation⁴⁷. A hallmark of Treg subversion is the expression of the Notch4 receptor⁴⁷. The here identified Treg/Th2 cells do not express Notch4, are Foxp3 positive (Supplementary Table 2) and show high expression of *Rora*, *Ikzf2*, and *Il2ra* (Fig. 1c and Supplementary Table 2) suggesting that these cells are stable, tissue adapted Tregs with repressive ability as previously described^{47, 48, 49, 50}. However, to fully dissect the biology of Treg/Th2 cells, lineage tracing experiments are warranted”.

Collectively, our data resolve the heterogeneity of murine lung pTh2 cells, reveal their similarities with previously overlooked lung Il1rl1-expressing Th cells (Treg/Th2), and highlight the need to define markers to distinguish the pTh2 cell subsets from the Treg/Th2 subset”.

And these sentences in the result on page 10 from line 457 to line 460:

*“In addition, we identify a unique Th subset that has features of both Treg and Th2 cells (Treg/Th2). Cells in this cluster express *Rora*, *Gata3* and *Il1rl1* (encoding ST2), which were identified as markers of tissue adaptation for Tregs⁴⁸. These cells also express high levels of *Ikzf2* which is linked to Treg stability^{47, 49}. These observations suggest that the majority of the Treg/Th2 cells are stable tissue adapted Tregs. Intriguingly, the pTh2 cell subsets show some similarities with the Treg/Th2 cell subset. Many marker genes previously attributed to lung pTh2 cells such as *Il1rl1* (ST2) and *Bhlhe40* are not unique to the pTh2 cell subsets”.*

- The explanation of Fig 7b/c is not convincing on IL-5 and IL-13. In WT IL-13 is much higher than HDAC cKO. Can the authors review the results?

Reply: The aim of Fig. 7b and Fig. 7c was to determine whether pTh2 cells from WT (Fig. 7b) and HDAC1-cKO (Fig. 7c) have a similar transcriptional signature. **Fig. 7b** compares WT pTh2 to WT Th2 on transcriptomic level, while **Fig. 7c** compares HDAC1-cKO Th2 to HDAC1-cKO pTh2 on transcriptomic level. Since a pathogenic program is already induced in HDAC1-KO Th2 cells due to the absence of HDAC1, the difference between HDAC1-cKO Th2 cells and HDAC1-cKO pTh2 cells might not be as high as the difference between WT Th2 and WT pTh2. Overall,

However, in **Extended Data Fig. 6a**, we compared HDAC1-cKO Th2 and WT Th2 (similar cell types from the different genotypes), and could show that both IL-5 and IL-13 were highly upregulated in the HDAC1-cKO Th2 cells. Similarly, we observed a strong upregulation of IL-5 in HDAC1-cKO pTh2 as compared to WT pTh2 cells (**Extended Data Fig. 6b**). These two figures demonstrate that, the loss

of HDAC1 predisposes the Th2 cells to acquire a pathogenic state and to express higher IL-5 and IL-13, and that the expression of IL-5 in pTh2 cells is strongly regulated by HDAC1.

For a better overview, we add the corresponding figure parts here in one **Revision Fig. 11**:

Fig. 7b,c

Extended Data Fig. 6a,b

Revision Fig.11| Comparisons between the different in vitro populations at transcriptomic level.

- Why wasn't IL-10 and AREG (Amphiregulin) shown in proteomics data? Can the authors explain?

Reply: The reviewer correctly observed that IL-10 and AREG are missing in our proteomic data (as also IL-5). These cytokines were not detected by mass spectrometry. One of the caveats of mass spectrometry is that not all peptides can be detected either due to their size or technical issues. However, we detected both IL-5 and IL-10 by flow cytometry (**Revision Fig. 8**). We could not find an appropriate antibody for flow cytometry against murine AREG to validate the expression of AREG by pTh2 cells.

- The final part on explaining the molecular mechanism is very shallow and weak. Epigenetic work on these cells need to take place to explain how HDAC1 control IL-5 and IL-13 loci without changing GATA3.

Reply: As highlighted above, we performed ATAC-seq of WT and HDAC1-cKO Th2 and pTh2 cells cultured for 24 and 48 hours after activation to enable us to shed some light on the potential mechanisms regulating the differentiation of pTh2 cells. We have now added a new figure (Fig. 9) to address this issue.

- The peTh2 cells are bad per authors definition, do they differ in their work in transfer exp with parasitic infection from regular Th2 and TmTh2? Will they not act or are they only super stimulated cells?

Reply: In an allergic reaction, peTh2 will have a detrimental function, as they produce high levels of IL-5, leading to exacerbation of the disease. However, this reviewer is right, in a parasitic infection like the helminth *Heligmosomoides polygyrus*, these cells are crucial as reduced IL-5 production was associated in impaired protection (PMID: 31900338). The definition or terminology "pathogenic" Th2 was defined by groups with important studies in the field of allergy, and is found in recognized reviews (PMID: 33845475, PMID: 36016937, PMID: 36003397), and we took over this definition. We agree with this reviewer that this pathogenicity applies to allergic diseases, as these inflammatory Th2 cells contribute strongly to the pathogenesis of the disease but this does not apply to the context of parasitic infection.

Minor Comments

- "Ref" by references 9,10,18)

Reply: Our referencing is in accordance with the journal's style.

- Authors should show FACS plots for KLRG1 and CD27 in Ext. Fig4

Reply: We included them.

- Anti-GITR blocking antibodies is very confusing, please rephrase it.

Reply: This statement is not in the manuscript.

- **Can the author explain briefly how they normalized 2 different RNA-seq for their comparison in 7 d,e or was the comparison only on gene numbers?**

Reply: We used all the differentially expressed genes in cluster 2 and compared them to the differentially expressed genes between the pTh2 and Th2 cells.

- **FACS plots should be unified in their shape between all figures main and extended.**

Reply: We have updated the FACS plots as appropriate.

V1

REVIEWERS' COMMENTS

Reviewer #1 (Remarks to the Author):

The authors have addressed my comments. I have no further comments.

Reply: We thank this reviewer for providing insightful and constructive comments that allowed us to improve our manuscript.

Reviewer #3 (Remarks to the Author):

no more comments

Reply: We thank this reviewer for providing insightful and constructive comments that allowed us to improve our manuscript.